# A ROBUST PPG FOUNDATION MODEL USING MULTI-MODAL PHYSIOLOGICAL SUPERVISION

## ABSTRACT

Photoplethysmography (PPG), a non-invasive measure of changes in blood volume, is widely used in both wearable devices and clinical settings. Although recent work has explored PPG foundation models using large-scale intensive care unit (ICU) datasets, these efforts often assume the need for clean and high-quality signals. In contrast, we argue that the inherent noise and variability in ICU datasets can be harnessed to build more robust and generalizable representations. To address this, we propose a PPG foundation model that leverages accompanying electrocardiogram and respiratory signals in ICU datasets to select contrastive samples during pretraining. Our approach allows the model to retain and learn from noisy PPG segments, improving robustness without requiring multimodal inputs at inference. Our model, pretrained on 3x fewer subjects than existing state-of-the-art approaches, achieves performance improvements of up to 36% in classification and 42% in regression on 14 out of 15 diverse downstream tasks, including stress and heart rate prediction. Our results demonstrate that multimodal supervision can leverage clinical data to enable the development of robust, unimodal foundation models for both clinical and consumer-level data.

## 1 INTRODUCTION

Wearable devices are rapidly emerging as powerful tools to monitor physiological and behavioral signals in everyday life. These devices typically rely on embedded sensors that must meet strict design constraints: they must be small, low-power, cost-effective, and unobtrusive. However, these constraints often compromise signal quality, introducing noise and variability that significantly challenge downstream tasks. Consequently, there is a critical need for robust models that can learn effective representations from noisy signals while maintaining high accuracy and generalizability across diverse conditions. Among the various biosignals used in wearables, photoplethysmography (PPG) has gained prominence due to its simplicity, low energy consumption, and compatibility with optically-based sensing hardware. PPG measures changes in blood volume in peripheral tissue, allowing the estimation of vital signs such as heart rate and blood pressure (Elgendi et al., 2019). Unlike electrocardiography (ECG), PPG is more prone to motion artifacts and signal noise (Fine et al., 2021). Nevertheless, PPG's ability to reflect vascular dynamics makes it a promising candidate for foundation models that aim to generalize across multiple tasks and conditions.

The advent of PaPaGei (Pillai et al., 2024), the first open-source PPG foundation model, highlights the growing interest in building general-purpose representations from wearable biosignals. PaPaGei demonstrates substantial performance gains over engineered features across multiple downstream tasks, including hypertension classification, blood pressure estimation, and heart rate prediction. These results suggest that foundation models can capture rich, transferable representations of PPG signals. However, PaPaGei depends on extensive preprocessing to extract clean morphological features for pretraining, and other existing approaches often rely on proprietary datasets (Saha et al., 2025; Abbaspourazad et al., 2023), limiting reproducibility and/or scalability. In this work, we investigate whether the reliance on curated, denoised PPG signals can be relaxed by leveraging co-recorded multimodal signals from large-scale intensive care unit (ICU) datasets. Specifically, we propose a novel PPG foundation model that uses co-recorded high-quality signals (electrocardiogram (ECG) and respiratory (RESP) data) to select contrastive PPG samples during pretraining. This allows us to learn from relatively noisy clinical PPG data, without requiring explicit denoising or morphological feature extraction. Importantly, our model requires only PPG signals at infer-

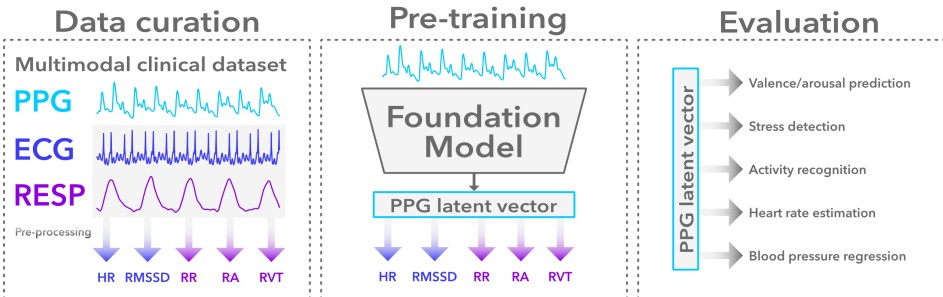

Figure 1: **Multimodal contrastive supervision framework.** (Left) The electrocardiogram (ECG) and respiratory (RESP) data co-recorded with PPG is segmented into 10s windows. Five metrics are extracted from the ECG and RESP segments that summarize those windows in a 5-dimensional vector. (Middle) The metrics are used to generative contrastive samples during pretraining. (Right) The unimodal PPG embeddings are evaluated using various tasks for unseen datasets.

ence, with multimodal data used exclusively during pretraining to enhance representation learning, as shown in Figure 1. Our key contributions are as follows:

- We demonstrate that leveraging open-source, co-recorded multimodal signals (ECG and RESP) from ICU datasets to guide PPG foundation model pretraining significantly enhances performance across a diverse set of downstream tasks.

- We introduce within-subject linear probing as a complementary evaluation method for PPG foundation models, enabling a more detailed and subject-specific assessment of representation quality and generalization beyond standard cross-subject metrics.

- Our approach outperforms PaPaGei on all but one downstream task, particularly on field-like datasets, despite using a significantly smaller subject pool and a single pretraining dataset. This highlights the efficiency, scalability, and robustness of our method.

## 2 RELATED WORK

Foundation models have demonstrated strong generalization across a wide range of domains, driven by large-scale pretraining and self-supervised learning (Bommasani, 2021). These models are often trained using self-supervised learning techniques that involve generating masked or incomplete data (Devlin et al., 2019). Generative and/or predictive pretraining has been replicated with success in other fields as well, including computer vision (He et al., 2022), pretraining for timeseries (Nie et al., 2022), and biosignals (Kostas et al., 2021; Chen et al., 2021; Chien et al., 2022; Dong et al., 2023; Liu et al., 2023; Yun et al., 2024; Zhang et al., 2024; Geenjaar & Lu, 2025). An important downside to these methods is that they can still be sensitive to noise. In practice, biosignals like PPG are far noisier than image or text data, and can be highly subject-dependent, influenced by factors such as skin tone and body composition (Bent et al., 2020). Generative approaches may thus fail to capture good embeddings for high-noise segments. Predictive approaches like JEPA (LeCun, 2022) aim to mitigate this by learning abstract representations without full reconstruction. Still, they remain sensitive to slow-varying or predictive noise patterns (Sobal et al., 2022). This slow and/or predictive noise may be induced by movement in PPG data. JEPA approaches may thus still struggle to learn good embeddings from noisy PPG data. Other, nongenerative or predictive self-supervised approaches have also been proposed for PPG data, including motif matching (Xu et al., 2023; 2024; Saha et al., 2025), morphology-based contrastive learning (Pillai et al., 2024), and temporal- or participant-based contrastive learning (Tonekaboni et al., 2021; Abbaspourazad et al., 2023). Although these approaches do not rely on generation and/or prediction during pretraining, they may still be sensitive to noise. For instance, PaPaGei relies on morphological features that are difficult to extract accurately from noisy segments, which can result in poor supervision or the exclusion of data. Moreover, similar to JEPA approaches, time- or participant-based contrastive learning that uses two noisy positive pairs may learn to focus on slow-frequency features, as opposed to potentially important medium-frequency signals like systolic peaks in the PPG signal.

Unimodal self-supervised supervision is only one subset of foundation models for general representation learning (Li et al., 2024). Other subsets include label supervision, multimodal supervision, and multimodal fusion models (Li et al., 2024). Given that labels vary significantly between PPG datasets, and future inference may be restricted by training a PPG foundation model using a specific set of classes, it is not practical to use label supervision for PPG foundation models. Moreover, although multimodal biosignal foundation models can be made robust to modality dropout (Liu et al., 2023; Fang et al., 2024), it is preferable to train a PPG foundation model that only requires PPG as an input to make it as general for wearable use as possible. Specifically, we focus on models that can be trained with multimodal inputs but deployed using only a single modality, making them practical for all types of wearables. In fact, large-scale open-source PPG pretraining datasets are often clinical datasets, which record multiple modalities from patients almost by default. To harness the availability of this data, we therefore focus on multimodal supervision as a method to train a PPG foundation model. Multimodal supervision models for computer vision include CLIP (Radford et al., 2021; Jia et al., 2021), and work for biosignals include multimodal contrastive pretraining (Raghu et al., 2022), BioFAME (Liu et al., 2023), SleepFM (Thapa et al., 2024), cross-modal masked auto-encoding learning (Fang et al., 2024), and CiTrus (Geenjaar & Lu, 2025). All of these works include various signals that are not included in most wearables, such as ECG, EOG, or EEG. Specifically for PPG data, works that use multimodal supervision for foundation models include SensorLM (Zhang et al., 2025), and work that simultaneously uses PPG and accelerometer data (Abbaspourazad et al., 2024). Both models are trained on closed-source datasets. In contrast, our approach uses open-source ICU data and relies only on PPG at inference time. We leverage biosignals co-recorded with PPG to construct a physiologically grounded supervision signal during pretraining, enabling robust representation learning from noisy clinical PPG data while maintaining scalability and reproducibility. This allows the community to build, evaluate, and extend upon our method by training on the same set of data. In fact, we share a list of the exact data files we use for our pretraining dataset in the Supplementary Material.

## 3 METHODS

Let $\mathcal{D} = \{(\text{PPG}, \text{ECG}, \text{RESP})^{n,s}\}_{n=1...N, s=1...S}$ denote the multimodal biosignal dataset, where N is the number of subjects and $S$ is the number of sessions per subject. Each tuple $(\text{PPG}, \text{ECG}, \text{RESP})^{n,s}$ contains time-aligned signals from each modality, sampled at the same frequency and of equal length. The continuous signals are segmented into non-overlapping 10-second windows following PaPaGei (Pillai et al., 2024), $\mathbf{w}_t^{n,s} = (\text{PPG}, \text{ECG}, \text{RESP})_{t:(t+10s)}^{n,s}$. The PPG segment $\mathbf{x}_t^{n,s} = \mathbf{w}_{t,\text{PPG}}^{n,s}$ is used as the model input, while the corresponding ECG and RESP segments are utilized to compute physiological metrics that guide contrastive supervision.

### 3.1 A PHYSIOLOGICAL METRIC SPACE FOR CONTRASTIVE SUPERVISION

To ensure our PPG foundation model is robust to naturally occurring noise, such noise must be well-represented in its pretraining dataset. Previous work relied on morphological features directly extracted from PPG segments to define contrastive targets (Pillai et al., 2024). Noisy segments, which are important to include in the pretraining dataset (Saha et al., 2025), consequently either need to be discarded or lead to inaccurate contrastive targets during pretraining. By contrast, ECG and RESP signals are typically less noisy and provide unique and physiologically relevant information. We exploit this robustness to derive a continuous physiological metric space that reflect the underlying cardio-respiratory state of the subject during each 10-second segment. Moreover, by precomputing metrics from the ECG and RESP data, we can ensure metrics do not fall outside of known physiological ranges, and filter the metrics to ensure that any noise in the ECG and RESP data does not significantly affect our constructed physiological metric space.

**ECG-derived metrics.** We extract two cardiovascular targets from each 10s ECG waveform: heart rate (**HR**) and the root mean square of successive differences (**RMSSD**). These are metrics that have relatively good repeatability for short segments (Schroeder et al., 2004; Nussinovitch et al., 2011; Shaffer & Ginsberg, 2017). Both provide a measure of heart rate variability (HRV), whereas resting HR is an indicator of all-around fitness and even cardiovascular disease (Fox et al., 2007), RMSSD is sensitive to autonomic function and stress (Kim et al., 2018). Especially because we are able to

filter the derived metrics before using them as contrastive targets, they are less sensitive to peripheral noise than PPG and serve as a physiological target for cardiac dynamics.

**RESP-derived metrics.** We compute three respiratory features from each 10s RESP waveform: respiratory rate (**RR**), respiratory amplitude (**RA**), and respiratory volume per time (**RVT**). These metrics reflect different aspects of breathing behavior, such as rhythm and tidal volume, and can indicate stress or enhanced attention (Widjaja et al., 2013), or disorders (Brinkman et al., 2018).

By precomputing and filtering these metrics to ensure physiological plausibility and reduce noise-induced artifacts, we obtain a stable, multidimensional metric space. Contrastive relationships are then defined based on similarity in this space rather than using potentially noisy PPG morphology. This design enables the model to learn representations that are better aligned with meaningful physiological variation, and more robust to naturally occurring PPG noise.

**Pretraining setup and learning objective.** Given a batch of B PPG segments, two augmented views are generated for each segment, yielding 2B inputs. These are encoded using a shared convolutional neural network $f_\theta(\cdot)$, producing embeddings $\{\mathbf{v}_i \in \mathbb{R}^{512}\}_{i=1\ldots 2B}$. Each embedding $\mathbf{v}_i$ is associated with a physiological metric vector $\mathbf{y}_i$. For each anchor embedding $\mathbf{v}_i$, the other embeddings $\mathbf{v}_j$ ($j \neq i$) are ranked according to the distance between their physiological targets $d(\mathbf{y}_i, \mathbf{y}_j)$ in the metric space. Embeddings corresponding to more physiologically similar segments are ranked higher. Formally, define the set $\mathcal{S}_{i,j} = \{\mathbf{v}_k \mid k \neq i,\ d(\mathbf{y}_i, \mathbf{y}_k) \geq d(\mathbf{y}_i, \mathbf{y}_j)\}$, which contains all embeddings that are further away than $\mathbf{v}_j$ is from $\mathbf{v}_i$. Then, we employ the rank-n-contrast (RNC) loss (Zha et al., 2023), which encourages embeddings that are closer in the physiological metric space to be closer in the learned representation space as well:

$$\mathcal{L}_{\text{RNC}} = \frac{1}{2B} \sum_{i=1}^{2B} \frac{1}{2B-1} \sum_{\substack{j=1 \\ j \neq i}}^{2B} - \log \frac{\exp(\text{sim}(\mathbf{v}_i, \mathbf{v}_j)/\tau)}{\sum_{\mathbf{v}_k \in \mathcal{S}_{i,j}} \exp(\text{sim}(\mathbf{v}_i, \mathbf{v}_k)/\tau)} \tag{1}$$

where $\text{sim}(\cdot, \cdot)$ denotes cosine similarity and $\tau$ is a temperature hyperparameter. This loss anchors the learned PPG embeddings to the robust physiological metric space derived from multimodal signals, improving noise robustness and encouraging physiologically meaningful representations.

# 4 EXPERIMENTAL SETUP

**Datasets.** For pretraining, we use the MIMIC-III Waveform Database Matched Subset (Goldberger et al., 2000; Johnson et al., 2016; Moody et al., 2020)[1], which contains waveform data from 10,282 ICU patients. We selected this dataset among the three used by PaPaGei because its subjects are neither asleep nor under anesthesia, unlike the other two datasets. This allows patients to move their arms naturally, introducing realistic movement artifacts and noise that improve the robustness of our model. The dataset includes multiple time-aligned

Table 1: **Downstream dataset information**. More information is provided in Appendix B. # P is short for number of participants, # S is short for number of total samples.

| Datasets | Task | Task type | # P | # S |
|---|---|---|---|---|
| WESAD | Stress | Clf (2) | 15 | 4125 |
| (Schmidt et al., 2018) | Affect | Clf (4) | 15 | 4125 |
| PPG-DaLiA | Daily activities | Clf (9) | 15 | 12865 |
| (Reiss et al., 2019) | Heart rate (HR) | Reg | 15 | 64697 |
| EEVR | Valence | Clf (2) | 37 | 10508 |
| (Singh et al., 2024) | Arousal | Clf (2) | 37 | 10508 |
| PPG-BP | Hypertension | Clf (2) | 219 | 657 |
| | Average HR | Reg | 219 | 657 |
| | Systolic BP | Reg | 219 | 657 |
| (Liang et al., 2018) | Diastolic BP | Reg | 219 | 657 |
| VitalVideos | Systolic BP | Reg | 100 | 300 |
| (Toye, 2023) | Diastolic BP | Reg | 100 | 300 |
| WildPPG | HR (green) | Reg | 64 | 304708 |
| | HR (infrared) | Reg | 64 | 304708 |
| (Meier et al., 2024) | HR (red) | Reg | 64 | 304708 |

biosignals sampled at 125Hz. The pretraining data preprocessing pipeline, detailed in Appendix A, filters the data to retain 4,998 subjects, yielding approximately 20 million 10-second PPG segments (about 56,000 hours of data). We evaluate our model on unseen datasets and tasks, an overview of

---

[1] https://physionet.org/content/mimic3wdb-matched/1.0/

these datasets is given in Table 1. In our selection of downstream datasets we focus on wearable-level data to verify the robustness of our model to noise in the PPG signal. Specifically, PPG-BP and VitalVideos are similar to clinical-level PPG data, WESAD and EEVR are lab environment datasets, and DaLiA and WildPPG are field-like datasets, which exhibit the highest noise levels.

**Backbone & pretraining.** We use a 1D ResNet-26 convolutional encoder [2] $f_\theta(\cdot)$ with instance normalization applied to the input. The network comprises 12 residual blocks, each using a kernel size of 11 and stride 2. The initial convolution outputs 128 filters, doubling every four layers. Spatial resolution is downsampled by a factor of 2 every two layers via max-pooling. Each unfiltered 10-second PPG window $\mathbf{x}^{n,t}$ is passed through the network to produce embeddings used in our contrastive learning objective (Eq. 1). For data augmentation, two random transformations are applied to each input window, selected from: GaussianNoise (p = 0.25), Negation (p = 0.20), Scaling (p = 0.40), and RandomCrop (p = 0.50). These augmentations follow the same strategy as PaPaGei (Pillai et al., 2024). Additional hyperparameter details are provided in Appendix C.

**Evaluation across subjects.** We evaluate our foundation model on 15 downstream tasks from six unseen datasets, encompassing both classification and regression problems. Classification tasks include stress, affect, arousal, valence, activity, and hypertension detection, while regression tasks cover heart rate prediction in field, daily activity, and clinical settings, as well as diastolic and systolic blood pressure estimation. Table 1 provides a detailed list of datasets and tasks, with more information in Appendix B.

Following PaPaGei (Pillai et al., 2024), we assess representation quality and generalizability using linear probing. Linear probing measures linear predictability from inferred embeddings while keeping the backbone weights frozen. For classification tasks, we use logistic regression, and for regression we use ridge regression. Hyperparameters for both models are tuned via 5-fold cross-validation on the training and validation splits, and are discussed in Appendix C. Both probes are implemented using scikit-learn (Pedregosa et al., 2011). Final results are averaged over five test folds. Model selection uses macro F1 score for classification and mean absolute error (MAE) for regression. Additionally, we report accuracy (ACC) and area under the receiver operating characteristic curve (AUC) for classification, as well as mean squared error (MSE) and mean absolute percentage error (MAPE) for regression.

**Evaluation within subjects.** To better assess model performance in realistic deployment scenarios, we introduce a within-subject linear probing evaluation protocol. Wearable devices are typically used by individual users, and it is often feasible to obtain labeled segments over time through user interaction or automatic annotation. Since physiological patterns can vary significantly across individuals, evaluating linear probe performance separately for each subject provides insight into the model's ability to generalize under subject-specific distributions. This evaluation uses the same linear probe architecture, hyperparameter tuning strategy, and metrics as in across-subject evaluation. However, instead of k-fold cross-validation across subjects, each fold corresponds to a user. Train-

---

[2] https://github.com/hsd1503/resnet1d

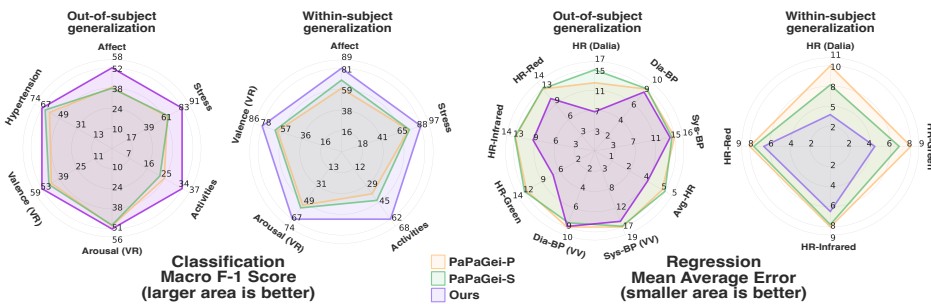

Figure 2: **Comparison with state-of-the-art**. (Left) Classification results in terms of their macro F-1 score (**larger area is better**). (Right) Regression results in terms of their mean average error (**smaller area is better**). We evaluate across subject linear probing, and within subject linear probing.

Table 2: **Downstream across-subject linear probing results.** Results are averaged over 5 test folds; standard deviations are reported in Appendix D. For classification tasks, higher values indicate better performance, measured by macro F1 score (MF1), accuracy (ACC), and area under the receiver operating characteristic curve (AUC). For regression tasks, lower values are better, evaluated using mean absolute error (MAE), mean squared error (MSE), and mean absolute percentage error (MAPE).

| | PaPaGei-P | | | PaPaGei-S | | | Ours | | |
|---|---|---|---|---|---|---|---|---|---|
| Clf (↑) | **MF1** | **ACC** | **AUC** | **MF1** | **ACC** | **AUC** | **MF1** | **ACC** | **AUC** |
| Stress | 0.65 | 0.79 | 0.8 | 0.67 | 0.78 | 0.76 | 0.83 | 0.88 | 0.93 |
| Affect | 0.4 | 0.49 | 0.69 | 0.39 | 0.47 | 0.66 | 0.52 | 0.6 | 0.78 |
| Activities | 0.25 | 0.33 | 0.72 | 0.23 | 0.29 | 0.68 | 0.34 | 0.39 | 0.8 |
| Arousal | 0.49 | 0.56 | 0.53 | 0.49 | 0.54 | 0.51 | 0.51 | 0.55 | 0.54 |
| Valence | 0.46 | 0.63 | 0.55 | 0.48 | 0.62 | 0.54 | 0.53 | 0.61 | 0.58 |
| Hypertension | 0.6 | 0.65 | 0.64 | 0.64 | 0.69 | 0.71 | 0.67 | 0.71 | 0.74 |
| Avg | 0.48 | 0.58 | 0.65 | 0.48 | 0.56 | 0.64 | **0.57** | **0.63** | **0.73** |
| Reg (↓) | **MAE** | **MSE** | **MAPE** | **MAE** | **MSE** | **MAPE** | **MAE** | **MSE** | **MAPE** |
| HR (Dalia) | 12.7 | 303 | 0.14 | 15.1 | 407 | 0.17 | 7.3 | 139 | 0.08 |
| Avg-HR | 4.76 | 40.0 | 0.07 | 4.95 | 43.4 | 0.07 | 3.72 | 23.8 | 0.05 |
| Sys-BP | 14.9 | 366 | 0.12 | 14.4 | 351 | 0.12 | 14.1 | 326 | 0.11 |
| Dia-BP | 8.73 | 121 | 0.12 | 8.77 | 122 | 0.12 | 8.3 | 114 | 0.12 |
| Sys-BP (VV) | 17.0 | 516 | 0.13 | 16.8 | 500 | 0.13 | 15.8 | 472 | 0.12 |
| Dia-BP (VV) | 8.7 | 143 | 0.11 | 8.22 | 125 | 0.1 | 8.63 | 138 | 0.11 |
| HR-Green | 12.2 | 266 | 0.17 | 12.5 | 273 | 0.17 | 7.41 | 149 | 0.1 |
| HR-Infrared | 12.6 | 273 | 0.17 | 12.7 | 277 | 0.17 | 9.81 | 212 | 0.14 |
| HR-Red | 12.7 | 279 | 0.17 | 12.7 | 284 | 0.17 | 10.7 | 233 | 0.15 |
| Avg | 11.6 | 256 | 0.13 | 11.8 | 265 | 0.14 | **9.53** | **201** | **0.11** |

ing, validation, and test sets are computed based on the temporally varying targets of that user, and results are averaged across 5 randomly selected users, more information is provided in Appendix C. Only datasets with temporally varying targets for each subject are included (e.g., affect detection, heart rate), while datasets with static per-subject labels, such as PPG-BP, are excluded. This leaves the 9 tasks from the WESAD, DaLiA, EEVR, and WildPPG datasets.

**Baselines.** We compare against PaPaGei-S and PaPaGei-P, the current state-of-the-art open-source PPG foundation models. PaPaGei-S uses a morphology-based contrastive learning framework, wherein morphological features are extracted from the raw PPG signal and used to construct positive and negative sample pairs during pretraining (Pillai et al., 2024). In contrast, PaPaGei-P employs a subject-aware contrastive loss, generating positive pairs from segments of the same individual and negatives from different individuals.

## 5 RESULTS

Our model outperforms both PaPaGei-S and PaPaGei-P on nearly all (14 out of 15) downstream tasks in across-subject and (9 out of 9) within-subject linear probing evaluations, as shown in Figure 2. Notably, these gains are achieved despite using 3x fewer pretraining subjects than PaPaGei, demonstrating the efficiency and robustness of our approach. Detailed numerical results for the across-subject evaluation are provided in Table 2, while within-subject results are summarized in Table 3. These results highlight the effectiveness and versatility of our model across diverse PPG analysis scenarios.

**Evaluation across subjects.** For across-subject evaluation, our model achieves consistent and substantial improvements over prior approaches. Notable classification gains include stress (0.83 vs. 0.67), affect detection (0.52 vs. 0.4), and daily activity classification (0.34 vs. 0.25) reflecting stronger generalization across user states and behaviors. Regression tasks show large improvements both under field-like (DaLiA: 7.3 vs. 12.7, WildPPG (Green): 7.41 vs. 12.2) and clinical conditions (PPG-BP: 3.72 vs. 4.76). Even on tasks where PaPaGei specifically excels, such as systolic blood pressure regression, our model outperforms both PaPaGei variants on the PPG-BP (14.1 vs. 14.4) and VitalVideos (15.8 vs. 16.8) datasets. The only task our model underperforms PaPaGei-S at is diastolic blood pressure regression for the VitalVideos dataset (8.63 vs. 8.22). This may be due to

Table 3: **Within-subject downstream linear probing results.** Results are averaged over 5 test folds, with standard deviations reported in Appendix D. For classification tasks, higher values indicate better performance; metrics include macro F1 score (MF1), accuracy (ACC), and area under the ROC curve (AUC). For regression tasks, lower values are better; metrics reported are mean absolute error (MAE), mean squared error (MSE), and mean absolute percentage error (MAPE).

| | PaPaGei-P | | | PaPaGei-S | | | Ours | | |
|---|---|---|---|---|---|---|---|---|---|
| Clf (↑) | **MF1** | **ACC** | **AUC** | **MF1** | **ACC** | **AUC** | **MF1** | **ACC** | **AUC** |
| Stress | 0.76 | 0.83 | 0.76 | 0.75 | 0.81 | 0.77 | 0.88 | 0.91 | 0.96 |
| Affect | 0.69 | 0.73 | 0.83 | 0.61 | 0.66 | 0.83 | 0.81 | 0.84 | 0.94 |
| Activities | 0.45 | 0.51 | 0.82 | 0.39 | 0.44 | 0.75 | 0.62 | 0.62 | 0.9 |
| Arousal | 0.56 | 0.59 | 0.61 | 0.54 | 0.56 | 0.58 | 0.67 | 0.69 | 0.78 |
| Valence | 0.65 | 0.69 | 0.69 | 0.63 | 0.67 | 0.66 | 0.78 | 0.8 | 0.86 |
| Avg | 0.62 | 0.67 | 0.74 | 0.58 | 0.63 | 0.72 | **0.75** | **0.77** | **0.89** |
| Reg (↓) | **MAE** | **MSE** | **MAPE** | **MAE** | **MSE** | **MAPE** | **MAE** | **MSE** | **MAPE** |
| HR (Dalia) | 7.92 | 121 | 0.09 | 10.4 | 199 | 0.12 | 4.09 | 41.2 | 0.05 |
| HR-Green | 7.22 | 122 | 0.09 | 8.45 | 186 | 0.11 | 4.67 | 73.6 | 0.06 |
| HR-Infrared | 8.03 | 142 | 0.11 | 8.27 | 151 | 0.11 | 6.68 | 118 | 0.09 |
| HR-Red | 7.85 | 130 | 0.1 | 8.26 | 185 | 0.11 | 6.82 | 111 | 0.09 |
| Avg | 7.76 | 129 | 0.1 | 8.83 | 180 | 0.11 | **5.57** | **86.1** | **0.07** |

subtle morphological differences that PaPaGei-S explicitly targets via morphology-based contrastive pretraining. In contrast, our model emphasizes robustness to noise and broader physiological variation, which may trade off fine-grained waveform sensitivity in cleaner datasets. However, our model performs better on PPG-BP for the same task (8.3 vs. 8.77), and on average (8.48 vs. 8.495). Our results also replicate the general improvement of PaPaGei-S over PaPaGei-P as reported in the original PaPaGei paper (Pillai et al., 2024). Minor differences in absolute numbers in PaPaGei's paper stem from our use of K-fold cross-validation, which better captures subject variability compared to single-split setups (Geenjaar & Lu, 2025).

**Evaluation within subjects.** Similar to the across subjects results, we find that classification improvements within subjects are also higher for stress (0.88 vs. 0.76), affect (0.81 vs. 0.69), and daily activities classification (0.62 vs. 0.45). Additionally, percentage improvements relative to the best PaPaGei model for within subject valence detection are much higher than across subjects: 10% across subjects, and 20% within subjects. The increased percentage improvement highlights that our model is even more accurate at tracking certain affective states in individual subjects over time. Figure 3 shows a visualization of the embedding space for a single subject from the PPG-DaLiA dataset, highlighting the difference between our method and our replication of PaPaGei-S (labeled as 'Unimodal'). In the figure, our model's embedding space shows a clear gradient in terms of heart rate, whereas the other models do not. Moreover, since data availability for a new user may be sparse, in Figure 4 we show how our model significantly outperforms both PaPaGei-P and PaPaGei-S, even on 10% data, the average performance of our model is better than the best PaPaGei model on 100% of the data. Data is removed in a stratified manner from the training set.

**Unimodal vs. multimodal pretraining.** To verify that differences in performance are due to our use of multimodal contrastive guidance during pretraining, and not because of architectural and pretraining data differences,

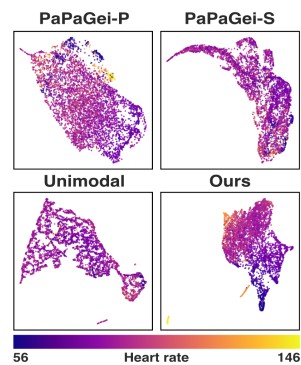

Figure 3: **UMAP plots(McInnes et al., 2018) colored by heart rate.** Data are from a single subject in the PPG-DaLiA dataset.

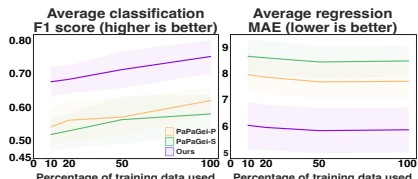

Figure 4: **Average performance across varying percentages of within-subject data.** Shaded areas represent standard deviation across folds.

we pretrain PaPaGei-S based on the available code [3] on our data. We adopt the PaPaGei-S pretraining objective, as it was shown to consistently outperform PaPaGei-P across downstream tasks in the original work. As a comparison, we match PaPaGei's backbone in our model, and use our proposed multimodal pretraining. All training hyperparameters are the same between the models, and are the same as the ones discussed in Appendix C. As detailed in Table 4, our multimodal pretraining consistently and substantially outperforms the unimodal PaPaGei-S baseline across all evaluated tasks, except diastolic blood pressure regression. These results strongly validate our core hypothesis: integrating complementary biosignal modalities during contrastive learning effectively mitigates the limitations inherent in unimodal morphology-based contrastive targets, leading to significantly enhanced robustness, generalization, and downstream task performance.

**Demographic analysis.** It is important to ensure that neither the specific set of subjects in the pretraining dataset nor the pretraining method lead to demographic biases in the model. To evaluate how bad demographic biases are, we use the systolic blood pressure regression task on the VitalVideos dataset, which records the Fitzpatrick skin tone (Gupta & Sharma, 2019), age, and sex of each subject. We use a leave-one-subject-out approach to perform linear probing, and also to select hyperparameters on the training set. The importance of skin tone in PPG analyses cannot be understated because PPG is an optical method, and skin tone can affect light wave reflectance (Fallow et al., 2013). Moreover, general device error for wearables recording heart rate have been found to be higher for darker skin tones (Gupta & Sharma, 2019). The results in Figure 5 show that some of the known biases appear in the models we tested. Specifically, performance is best for the lighest skin tone, and in case of skin tone 4 and 5 we see that our model performs worse than the PaPaGei-S model. In addition, performance for adults aged 47-61 is worst and performance is lower for female subjects. In the latter case, we observe that our model improves performance for both sexes. These findings underscore existing challenges in equitable biosignal modeling and highlight areas for future bias mitigation.

Table 4: **Unimodal vs multimodal pre-training, same architecture and data**. Results for both methods are averaged across 5 test folds, and standard deviations can be found in Appendix D. For the classification tasks, higher is better, and evaluation metrics are macro F-1 score (MF1), accuracy (ACC), and the area under the receiver operating characteristic (AUC). For regression tasks lower is better, and we use mean average error (MAE), mean squared error (MSE), and mean average percentage error (MAPE).

| | Unimodal pre-training | | | Multimodal pre-training | | |
|---|---|---|---|---|---|---|
| Clf (↑) | MF1 | ACC | AUC | MF1 | ACC | AUC |
| Stress | 0.63 | 0.79 | 0.84 | 0.76 | 0.83 | 0.86 |
| Affect | 0.39 | 0.51 | 0.71 | 0.43 | 0.53 | 0.71 |
| Activities | 0.19 | 0.32 | 0.7 | 0.31 | 0.37 | 0.78 |
| Arousal | 0.38 | 0.58 | 0.53 | 0.49 | 0.55 | 0.52 |
| Valence | 0.39 | 0.65 | 0.53 | 0.52 | 0.61 | 0.57 |
| Hypertension | 0.52 | 0.64 | 0.61 | 0.67 | 0.72 | 0.72 |
| Avg | 0.42 | 0.58 | 0.65 | **0.53** | **0.6** | **0.69** |
| Reg (↓) | MAE | MSE | MAPE | MAE | MSE | MAPE |
| HR (Dalia) | 16.0 | 443 | 0.18 | 8.25 | 167 | 0.09 |
| Avg-HR | 7.69 | 94.6 | 0.11 | 3.69 | 24.2 | 0.05 |
| Sys-BP | 15.8 | 408 | 0.13 | 14.1 | 326 | 0.11 |
| Dia-BP | 8.58 | 120 | 0.12 | 8.73 | 120 | 0.12 |
| Sys-BP (VV) | 16.9 | 505 | 0.13 | 16.2 | 467 | 0.12 |
| Dia-BP (VV) | 8.35 | 127 | 0.1 | 8.64 | 137 | 0.11 |
| HR-Green | 13.0 | 286 | 0.18 | 8.42 | 176 | 0.12 |
| HR-Infrared | 12.8 | 277 | 0.18 | 10.4 | 226 | 0.15 |
| HR-Red | 12.7 | 279 | 0.17 | 11.6 | 260 | 0.16 |
| Avg | 12.4 | 282 | 0.14 | **10.0** | **211** | **0.11** |

---

[3]https://github.com/Nokia-Bell-Labs/papagei-foundation-model

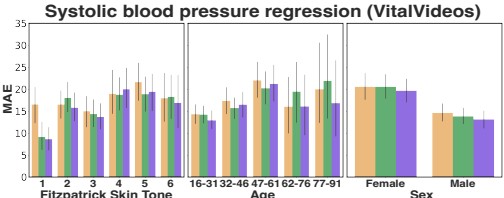

Figure 5: **Systolic blood pressure regression comparison across demographic variables.**

Figure 6: **WildPPG heart rate estimation comparison across PPG sensor location (x-axis) and type (y-axis).**

**Heart rate estimation ablations.** In Figure 6 we characterize how heart rate estimation performance varies across PPG recording locations and type of sensor (green, red, or infrared). We find that our model performs best across conditions, but especially for green wrist-worn PPG sensors. Further ablation compared with NeuroKit's (Makowski et al., 2021) automatic heart rate estimation tool is discussed in Appendix E.

**Backbone architecture ablation.** To understand the impact of the new architecture we use, which notably has a larger number of parameters (28.8M vs. 5-5.7M), we compare the PaPaGei backbone to our proposed backbone. As shown in Table 5, our architecture yields substantial performance gains across all tasks. These results suggest that our enhanced backbone and larger model capacity contribute to improved performance, indicating potential benefits from further scaling.

Table 5: **Architecture ablation, PaPaGei backbone architecture vs our proposed architecture, same pre-training**. Results for both methods are averaged across 5 test folds, and standard deviations can be found in Appendix D. For the classification tasks, higher is better, and evaluation metrics are macro F-1 score (MF1), accuracy (ACC), and the area under the receiver operating characteristic (AUC). For regression tasks lower is better, and we use mean average error (MAE), mean squared error (MSE), and mean average percentage error (MAPE).

| | PaPaGei Arch | | | Proposed Arch | | |
|---|---|---|---|---|---|---|
| Clf (↑) | **MF1** | **ACC** | **AUC** | **MF1** | **ACC** | **AUC** |
| Stress | 0.76 | 0.83 | 0.86 | 0.83 | 0.88 | 0.94 |
| Affect | 0.43 | 0.53 | 0.71 | 0.52 | 0.6 | 0.78 |
| Activities | 0.31 | 0.37 | 0.78 | 0.36 | 0.41 | 0.82 |
| Arousal | 0.49 | 0.55 | 0.52 | 0.53 | 0.57 | 0.57 |
| Valence | 0.52 | 0.61 | 0.57 | 0.53 | 0.62 | 0.58 |
| Hypertension | 0.67 | 0.72 | 0.72 | 0.71 | 0.75 | 0.77 |
| Avg | 0.53 | 0.6 | 0.69 | **0.58** | **0.64** | **0.74** |
| Reg (↓) | **MAE** | **MSE** | **MAPE** | **MAE** | **MSE** | **MAPE** |
| HR (Dalia) | 8.25 | 167 | 0.09 | 7.78 | 143 | 0.09 |
| Avg-HR | 3.69 | 24.2 | 0.05 | 3.8 | 26.3 | 0.05 |
| Sys-BP | 14.1 | 326 | 0.11 | 13.2 | 281 | 0.11 |
| Dia-BP | 8.73 | 120 | 0.12 | 8.16 | 109 | 0.12 |
| Sys-BP (VV) | 16.2 | 467 | 0.12 | 15.9 | 451 | 0.12 |
| Dia-BP (VV) | 8.64 | 137 | 0.11 | 8.04 | 123 | 0.1 |
| HR-Green | 8.42 | 176 | 0.12 | 7.61 | 149 | 0.1 |
| HR-Infrared | 10.4 | 226 | 0.15 | 9.82 | 206 | 0.14 |
| HR-Red | 11.6 | 260 | 0.16 | 10.8 | 231 | 0.15 |
| Avg | 10.0 | 211 | 0.11 | **9.45** | **191** | **0.11** |

## 6 CONCLUSION

This paper presents a PPG foundation model with robust multimodal pretraining, in which we use multiple biosignals alongside PPG to guide the contrastive PPG foundation model training. By using additional biosignals, we create accurate contrastive learning targets, allowing us to learn from relatively noisy clinical PPG data and improve performance on the downstream tasks. Through multiple ablation studies we demonstrate that our proposed pretraining approach greatly improves performance. In particular, our model outperforms both PaPaGei models, which are state-of-the-art PPG foundation models, in all but one out of 15 downstream tasks for across subject evaluations, and all downstream tasks for within subject evaluations. For the across subject classification and regression tasks we find improvements up to 36% for activity classification and up to 42% for field-like heart rate estimation, respectively. Notably, our model exhibits the largest performance gains for field-like datasets (DaLiA and WildPPG) and a lab dataset (WESAD). Given that these datasets focus on day-to-day PPG signals, we conclude that leveraging high-quality multimodal data during pretraining ensures our model is more robust to noise often seen in consumer-level data.

By using physiological metrics like heart and respiratory rate as contrastive learning targets, our method allows for a flexible combination of multiple biosignals. To further improve our current pretraining objective, one option is to include blood pressure as a target during pretraining, since it is often available in large-scale ICU datasets and is one of the tasks where our model performs most comparably to PaPaGei. Future work could also look at further improving low-data within subject performance by pretraining a model with subject-specific parameters. These parameters would then only need to be updated during a fine-tuning phase on a specific subject's data. Another important direction for future research is to reduce the effect of skin tone on PPG foundation model embeddings. In Figure 5 we analyze how systolic blood pressure regression performance differs across skin tones using the Fitzpatrick scale. However this scale doesn't fully capture skin tone diversity or the biases it creates in PPG recordings (Ware et al., 2020; Colvonen, 2021). It is thus important future work to include more diverse skin tones in the ptraining data, more comprehensively test for various demographic biases, and ultimately minimize these biases.

REPRODUCIBILITY STATEMENT

The code and weights for this paper are protected under a non-disclosure agreement (NDA) and can thus not be released to the public. To make the work as reproducible as possible, we have added a .txt file with the names of the pretraining files we use in the Supplementary Material. Moreover, both in the Methods section and Appendices A, B, and C we provide a very detailed explanation of our data curation, pre-processing, and experiment hyperparameters. Lastly, to make reproducing our results as easy as possible, we have used as many open-source implementations for our backbone architecture, contrastive training loss, and augmentations as possible.

ETHICS STATEMENT

Non-invasive health monitoring can revolutionize the healthcare system. It is important to ensure that groups of people can equally benefit from non-invasive health monitoring. To verify biases in systolic blood pressure regression, we perform a demographic analysis to understand what demographics our model is biased towards. Although this is a step in the right direction, it is important to develop methods that can counteract any biases and more thoroughly verify what biases exist in health foundation models. Further expansions of bias analyses across more demographics, and for more tasks is thus important, and the development of datasets that allow foundation models to be tested exhaustively before deployment is essential. The deployment of wearables for health monitoring is also accompanied by ethical and legal implications that must be addressed (Capulli et al., 2025). Finally, it is important that any health monitoring data from wearables is used **in alignment with a user's preferences**. We are committed to protecting participant and/or user privacy and welfare, and to ensuring scientific validity.

THE USE OF LARGE LANGUAGE MODELS (LLMS)

During the preparation of this manuscript, we did not substantially use LLMs. LLMs were only used to polish writing, and ensure the manuscript is approachable for a large audience. We have independently checked the correctness and clarity of the text.

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

## A    ECG AND RESP PRE-PROCESSING

We identify sessions containing more than one hour of continuous data across all three modalities: ECG, RESP, and PPG. The ECG and RESP signals are filtered using NeuroKit (Makowski et al., 2021), and then used to detect peaks: R-peaks in ECG and respiratory peaks/troughs in RESP. Signal regions without valid peaks are trimmed, and the remaining data is segmented into non-overlapping 10-second windows. From the RESP signal, we compute RR, RA, and RVT, and average them within each window. From the ECG signal, we extract HR and RMSSD. Any NaN values (i.e. in case a heart rate under 30 was observed) were linearly interpolated. Afterwards, all metrics are filtered using a low-pass filter with a cut-off frequency of 0.001Hz (since the metrics are sampled at 0.1Hz after averaging). The PPG signal is used in its raw, unfiltered form. During pretraining we map all of the metrics within the [0, 1] range to ensure that all metrics equally contribute to the distance computation. We decide lower and upper bounds for the range based on known physiological ranges, and by computing the lower and upper 4 standard deviations away from the mean across all metrics in the pretraining dataset. We land on the following ranges, which we use to map all metrics between [0, 1]: HR [30, 210], RMSSD [10, 200], RA [8, 60], RR [0, 2], and RVT [0, 0.88]. Any values outside of this range are clipped to be within the range. Lastly, during pretraining we select one session for each subject for pretraining, the names of the session files that correspond to files in the MIMIC Database can be found in the Supplementary Material.

## B  DOWNSTREAM DATASET INFORMATION

**WESAD**   The Wearable Stress and Affect Detection (WESAD) dataset (Schmidt et al., 2018) contains 15 subjects recorded in a lab setting. Although the dataset records data from a variety of physiological sensors, we only select the PPG data, which is recorded with a 64Hz sensor. In terms of PPG preprocessing we follow (Xu et al., 2023), whose preprocessing code is available on GitHub. We adapt the preprocessing code to obtain 10s non-overlapping segments, and we use PaPaGei's `resample_batch_signal` function to resample the segments to 125Hz to match the pretraining dataset. WESAD contains 4 classes: segments that consist of a baseline recording for each subject, segments where stress is induced using a Trier Social Stress Test (TSST) (Kirschbaum et al., 1993), segments where amusement is induced using a set of eleven funny video clips, and segments where subjects follow guided meditation. In case of our affect prediction task, we classify between each of these four classes. In case of the stress classification type, we classify between segments where stress is induced versus all other segments.

**PPG-DaLiA**   To better understand how well heart rate can be extracted from PPG during a wide range of activities under real-life conditions, (Reiss et al., 2019) introduced the PPG-DaLiA dataset, which records PPG at 64Hz. The dataset contains data from 15 subjects that perform eight different activities: (1) Sitting still for 10 minutes (2) Ascending/descending stairs for 5 min (3) Table soccer for 5 min (4) Cycling for 8 min (5) Driving a car for 15 min (6) Having a lunch break for 30 min (7) Walking for 10 min (8) Working for 20 min. During these daily activities, both a subject's PPG and ECG signals are recorded. The ECG signal is used as the ground truth for each 8 second window, with 2 second overlap between the windows. For the heart regression task, we use this label (`'label'`) because it is provided by the dataset and has been verified and preprocessed. Although the segment window is smaller than the pretraining dataset, both our and the PaPaGei backbone architecture can easily deal with slightly shorter segments because both use global averaging. For each subject, we first filter the PPG data with PapaGei's `preprocess_one_ppg_signal`, segment the data into 8 second windows with a 2 second overlap to match the target labels, and then use PaPaGei's `resample_batch_signal` function to resample the segments to 125Hz to match the pretraining dataset. The segments are z-scored for each subject. For the daily activities classification task, we use 10 second non-overlapping segments. The labels are sampled at 4Hz (`'activity'`), and we assign a label to a specific 10 second window if 75% or more of the window contains that specific activity. If there is no consensus on the window, we discard it. We first filter the PPG data with PapaGei's `preprocess_one_ppg_signal`, segment the data into 10 seconds non-overlapping segments to match the target labels, and then use PaPaGei's `resample_batch_signal` function to resample the segments to 125Hz to match the pretraining dataset.

**EEVR**   The Emotion Elicitation in Virtual Reality (EEVR) dataset (Singh et al., 2024) measures PPG data at 125Hz while 37 subjects are wearing a virtual reality (VR) headset. The study consists of baseline dataset collection, a VR familiarity task, and then a set of VR stimuli with post-exposure questionnaires. To evoke specific levels of arousal and valence, the authors use annotated 360° videos from a public database (Li et al., 2017), and select select videos based on four emotional quadrants of the Russell circumplex of affect (Russell, 1980). The circumplex contains two dimensions, valence and arousal, and the videos can thus be organized into high valence and low valence or high arousal and low arousal. The authors provide a csv file called `Raw_PPG.csv`. We use the `Participant_ID` column to separate data into specific subjects, and `Label_no_index` to separate each subject's session into a specific video with a high/low arousal and high/low valence label. Each video's corresponding PPG data is first filtered with PapaGei's `preprocess_one_ppg_signal`, and data is segmented into 10 second, 5 second overlapping windows. Each segment is labeled separately for arousal and valence, and since the PPG sampling rate matches that of our pretraining dataset, we do not resample the data.

**PPG-BP**   To better understand how PPG can be used to understand and predict cardiovascular disease, (Liang et al., 2018) released the PPG blood pressure dataset ( PPG-BP), with PPG sampled at 1000Hz. There are three PPG recordings for each subject that last around 2 second each, and 219 subjects in total. We noticed some issues with resampling the data, so we decided to linearly interpolate the data instead. Using `np.interp` (Harris et al., 2020), we interpolate each segment's frequency from 1000Hz down to 125Hz to match the pretraining dataset frequency. Given that

both our and PaPaGei's architecture use global averaging, both architectures can handle a variety of input sizes, so we didn't pad the input data, but before interpolation we did ensure the data was not longer than 2.1 seconds. Each subject has a recorded systolic blood pressure, diastolic blood pressure, and hypertension label. We use the same label for each of the three segments for a subject. After downsampling, we filter each segment with PapaGei's `preprocess_one_ppg_signal`, and z-score each segment.

**VitalVideos**   As an additional evaluation of blood pressure, we also evaluate our model on the VV-Small subset of the VitalVideos database. The demographics of this dataset are outlined on Page 6 of (Toye, 2023). The dataset contains systolic and diastolic blood pressure measurements for 100 subjects, and PPG sampled at 55-60Hz. Given that the sampling rate varies throughout recording, we interpolate the data to 125Hz, in order to match the sampling rate of the pretraining dataset, with `np.interp` (Harris et al., 2020) based on the provided sample timings. We also record the age, Fitzpatrick scale, and sex of each participant to perform our demographic analysis (See Figure 5). After interpolating, the PPG data for each subject is filtered with PapaGei's `preprocess_one_ppg_signal`, and data is segmented into non-overlapping 10 second windows. The PPG data is then z-scored for each subject. Yeah

**WildPPG**   To better understand how different placements of PPG sensors, different types (wavelengths) of PPG sensors, and daily activities impact heart rate estimation (Meier et al., 2024) released the WildPPG database. The dataset records data from 16 subjects, and each PPG sensor records at 128 Hz. The ground truth estimate of the heart rate is estimated with an ECG trace recorded from each subject's sternum. The dataset contains data for three types of PPG sensors: green, red, and infrared (IR), and four types of locations: wrist, head, ankle, and the sternum (chest in our manuscript). We follow the code provided by the authors on GitHub, but adapt the code in the following ways. We ensure that the ground truth heart rate from the ECG trace is estimated in 10s non-overlapping windows. Moreover, for each sensor, location, and subject, we filter the PPG data with PapaGei's `preprocess_one_ppg_signal`, segment the PPG data into non-overlapping 10s windows, and resample the segments from 128Hz to 125Hz to match the pretraining dataset with PaPaGei's `resample_batch_signal`. In case it is necessary, we trim the ground-truth heart rate segments based on the number of PPG windows. Then, we remove any segments where the ground-truth heart rate is zero (generally indicates that the heart rate could not be estimated), and then we z-score the PPG data for each sensor, location, and subject.

## C  HYPERPARAMETERS AND EXPERIMENTAL SETTINGS

**Architecture.** Our model's architecture is implemented in Py-Torch (Paszke et al., 2019), and consists of two main parts. First, the input to our model is a $(2 \times \text{batch\_size}, 1, 1250)$ tensor. We use 256 as the batch size during pretraining for all models. The reason we have twice as many segments along the batch dimension is because we sample two random augmentations, as described in the Methods section. This tensor first passes through an InstanceNorm1d layer, and then the 1D ResNet-26 architecture, as described in the Methods section of the main text. The hyperparameters for the ResNet are shown in Table 6. The output embedding for our model is thus a $(2 \times \text{batch\_size}, 512)$ tensor. During pretraining, we attach a linear layer to the gradient-detached embeddings. The gradient detaching is to ensure the backbone does not train to explicitly predict the metrics during pretraining. We use the predictions from the linear layer to monitor the model's training progress (i.e. metric predictions should get better during training). Moreover, after epochs 3-4 we see the metric predictions get worse, which we believe indicates overfitting. Hence, we do not consider checkpoints after the first 5 epochs.

Table 6: The 1D ResNet-26 hyperparameters

| | |
|---|---|
| in_channels | 128 |
| kernel_size | 11 |
| stride | 2 |
| groups | 1 |
| n_block | 12 |
| n_classes | 512 |
| downsample_gap | 2 |
| increasefilter_gap | 4 |
| use_bn | True |
| use_do | True |
| verbose | False |

**Checkpoint selection.** During pretraining we save checkpoints for the backbone every 5000 steps. To select the final checkpoint that we use for comparisons, we evaluate each checkpoint on the VitalVideos Systolic BP regression task. To ensure there is not data leakage, we use each model's training and validation set score during the hyperparameter selection process for the linear probe. The reason we use the VitalVideos Systolic BP regression task is because the results can be computed quickly. Evaluating every task for every checkpoint would require too much time and too many computational resources. In general, this metric will give us an idea about how well the model can still generalize to different datasets, and is thus valuable enough to select a checkpoint. For each model that we train in this paper we use a single checkpoint for all of the results.

**Linear probing K-folds.** There are generally two types of datasets. Datasets where each subject has a label. PPG-BP: average heart rate (Avg-HR), systolic blood pressure (Sys-BP), diastolic blood pressure (Dia-BP), and hypertension, and VitalVideos: systolic blood pressure (Sys-BP VV) and diastolic blood pressure (Dia-BP VV) from the VitalVideos dataset. The other type of dataset are datasets with labels that vary for each subject over time. WESAD: stress and affect, PPG-DaLiA: activities and heart rate, EEVR: arousal and valence, WildPPG: heart rate. For across subject linear probing, datasets where each subject has a label are stratified when computing 5 folds. Specifically, for regression tasks, values are binned into 10 bins, using an ordinal encoding, and based on quantiles in the dataset using `KBinsDiscretizer`. For classification tasks, no additional binning is necessary. The 5 splits are then obtained using `StratifiedKFold` with `random_state=42` and shuffling on. The training indices for each fold are split into training and validation indices with a training size of 0.75, `random_state=42`, shuffling on, and by stratifying the targets. For datasets where labels vary over time, we obtain 5 splits with `KFold`, `random_state=42`, and shuffling on. All names align with `scikit-learn`'s API (Pedregosa et al., 2011). For within subject linear probing, we first randomly shuffle all subjects with `np.random.default_rng(42)`, and select the subject that corresponds to the fold index. For regression tasks, we follow the same binning process described above to create a stratified test set. The test set is created using `train_test_split`, stratification, 0.2 as the test size, `random_state=42`, and shuffling on. The leftover data samples are then split into a stratified training and validation set with `train_test_split`, 0.8 as the training size, `random_state=42`, and shuffling on.

**Linear probing hyperparameters.** The search space includes $\alpha \in \{0.1, 1, 10, 100, 1000\}$ and `solver` $\in \{$`auto`, `cholesky`, `sparse_cg`$\}$. For regression tasks, we employ ridge regression with hyperparameters $C \in \{0.0, 0.1, 1, 10, 100\}$ and `max_iter` fixed to 10,000. The naming of these hyperparameters is aligned with the `scikit-learn` API (Pedregosa et al., 2011).

# D  STANDARD DEVIATIONS

Tables 7, 8, 9, and 10 report the standard deviations across 5 folds for the main tables in the text. Table 7 corresponds to Table 2 in the main text, and Table 8 corresponds to Table 3 in the main text. Tables 9 and 10 correspond to ablation Tables 4 and 5, respectively.

Table 7: **Downstream across subjects linear probing standard deviations across 5 folds**. Results for each method are averaged across 5 test folds, and standard deviations can be found in Appendix D. For the classification tasks, higher is better, and evaluation metrics are macro F-1 score (MF1), accuracy (ACC), and the area under the receiver operating characteristic (AUC). For regression tasks lower is better, and we use mean average error (MAE), mean squared error (MSE), and mean average percentage error (MAPE).

| | PaPaGei-P | | | PaPaGei-S | | | Ours | | |
|---|---|---|---|---|---|---|---|---|---|
| Clf (↑) | MF1 | ACC | AUC | MF1 | ACC | AUC | MF1 | ACC | AUC |
| Stress | 0.03 | 0.02 | 0.06 | 0.04 | 0.03 | 0.05 | 0.06 | 0.04 | 0.03 |
| Affect | 0.05 | 0.04 | 0.04 | 0.01 | 0.03 | 0.03 | 0.06 | 0.05 | 0.05 |
| Activities | 0.04 | 0.03 | 0.03 | 0.03 | 0.03 | 0.03 | 0.03 | 0.03 | 0.02 |
| Arousal | 0.02 | 0.01 | 0.02 | 0.02 | 0.02 | 0.02 | 0.03 | 0.02 | 0.04 |
| Valence | 0.02 | 0.01 | 0.04 | 0.02 | 0.02 | 0.02 | 0.02 | 0.04 | 0.03 |
| Hypertension | 0.03 | 0.03 | 0.04 | 0.03 | 0.04 | 0.05 | 0.06 | 0.05 | 0.07 |
| Avg | 0.03 | 0.03 | 0.04 | 0.03 | 0.03 | 0.03 | 0.04 | 0.04 | 0.04 |
| Reg (↓) | MAE | MSE | MAPE | MAE | MSE | MAPE | MAE | MSE | MAPE |
| HR (DaLiA) | 1.08 | 75.7 | 0.02 | 1.57 | 111 | 0.03 | 0.93 | 34.3 | 0.02 |
| Avg-HR | 0.27 | 6.04 | 0.0 | 0.35 | 3.56 | 0.0 | 0.24 | 4.56 | 0.0 |
| Sys-BP | 0.8 | 57.0 | 0.01 | 0.62 | 47.5 | 0.01 | 1.62 | 73.1 | 0.02 |
| Dia-BP | 0.34 | 6.61 | 0.01 | 0.31 | 6.44 | 0.01 | 0.55 | 11.9 | 0.01 |
| Sys-BP (VV) | 2.0 | 148 | 0.01 | 2.18 | 179 | 0.02 | 2.66 | 169 | 0.02 |
| Dia-BP (VV) | 1.1 | 56.5 | 0.01 | 1.35 | 53.7 | 0.02 | 1.37 | 64.0 | 0.02 |
| HR-Green | 1.51 | 68.1 | 0.04 | 1.28 | 54.4 | 0.03 | 1.36 | 53.7 | 0.03 |
| HR-Infrared | 2.05 | 86.1 | 0.04 | 2.17 | 89.5 | 0.05 | 2.08 | 91.0 | 0.04 |
| HR-Red | 0.79 | 29.7 | 0.01 | 0.77 | 27.4 | 0.01 | 0.41 | 11.9 | 0.01 |
| Avg | 1.1 | 59.3 | 0.02 | 1.18 | 63.7 | 0.02 | 1.25 | 57.1 | 0.02 |

Table 8: **Downstream within subjects linear probing standard deviations across 5 folds**. Results for each method are averaged across 5 test folds, and standard deviations can be found in Appendix D. For the classification tasks, higher is better, and evaluation metrics are macro F-1 score (MF1), accuracy (ACC), and the area under the receiver operating characteristic (AUC). For regression tasks lower is better, and we use mean average error (MAE), mean squared error (MSE), and mean average percentage error (MAPE).

| | PaPaGei-P | | | PaPaGei-S | | | Ours | | |
|---|---|---|---|---|---|---|---|---|---|
| Clf (↑) | MF1 | ACC | AUC | MF1 | ACC | AUC | MF1 | ACC | AUC |
| Stress | 0.04 | 0.03 | 0.08 | 0.1 | 0.08 | 0.12 | 0.09 | 0.07 | 0.04 |
| Affect | 0.09 | 0.08 | 0.09 | 0.13 | 0.11 | 0.08 | 0.11 | 0.1 | 0.04 |
| Activities | 0.14 | 0.13 | 0.05 | 0.14 | 0.14 | 0.09 | 0.11 | 0.13 | 0.04 |
| Arousal | 0.02 | 0.02 | 0.03 | 0.06 | 0.06 | 0.07 | 0.02 | 0.02 | 0.03 |
| Valence | 0.08 | 0.07 | 0.1 | 0.11 | 0.12 | 0.14 | 0.07 | 0.07 | 0.08 |
| Avg | 0.07 | 0.07 | 0.07 | 0.11 | 0.1 | 0.1 | 0.08 | 0.08 | 0.05 |
| Reg (↓) | MAE | MSE | MAPE | MAE | MSE | MAPE | MAE | MSE | MAPE |
| HR (DaLiA) | 0.88 | 26.7 | 0.01 | 1.03 | 40.0 | 0.01 | 1.02 | 18.9 | 0.02 |
| HR-Green | 0.65 | 36.0 | 0.01 | 0.38 | 48.7 | 0.01 | 1.02 | 36.6 | 0.02 |
| HR-Infrared | 1.5 | 69.0 | 0.03 | 1.54 | 80.5 | 0.03 | 1.71 | 71.6 | 0.03 |
| HR-Red | 0.44 | 28.6 | 0.01 | 0.5 | 45.8 | 0.01 | 1.06 | 37.4 | 0.02 |
| Avg | 0.87 | 40.1 | 0.02 | 0.86 | 53.7 | 0.02 | 1.2 | 41.1 | 0.02 |

Table 9: **Unimodal vs multimodal pre-training standard deviations across 5 folds**. Results for both methods are averaged across 5 test folds, and standard deviations can be found in Appendix D. For the classification tasks, higher is better, and evaluation metrics are macro F-1 score (MF1), accuracy (ACC), and the area under the receiver operating characteristic (AUC). For regression tasks lower is better, and we use mean average error (MAE), mean squared error (MSE), and mean average percentage error (MAPE).

| | Unimodal pre-training | | | Multimodal pre-training | | |
|---|---|---|---|---|---|---|
| Clf (↑) | MF1 | ACC | AUC | MF1 | ACC | AUC |
| Stress | 0.06 | 0.02 | 0.03 | 0.04 | 0.02 | 0.03 |
| Affect | 0.05 | 0.03 | 0.06 | 0.06 | 0.05 | 0.04 |
| Activities | 0.04 | 0.04 | 0.04 | 0.05 | 0.04 | 0.03 |
| Arousal | 0.02 | 0.02 | 0.02 | 0.02 | 0.02 | 0.02 |
| Valence | 0.0 | 0.0 | 0.02 | 0.01 | 0.03 | 0.02 |
| Hypertension | 0.05 | 0.04 | 0.04 | 0.02 | 0.02 | 0.03 |
| Avg | 0.04 | 0.02 | 0.03 | 0.03 | 0.03 | 0.03 |
| Reg (↓) | MAE | MSE | MAPE | MAE | MSE | MAPE |
| HR (DaLiA) | 2.64 | 167 | 0.04 | 1.0 | 36.8 | 0.03 |
| Avg-HR | 0.42 | 11.4 | 0.01 | 0.27 | 5.86 | 0.0 |
| Sys-BP | 0.37 | 33.2 | 0.0 | 0.54 | 50.8 | 0.01 |
| Dia-BP | 0.23 | 6.69 | 0.01 | 0.53 | 12.1 | 0.01 |
| Sys-BP (VV) | 1.76 | 142 | 0.01 | 1.45 | 98.9 | 0.01 |
| Dia-BP (VV) | 1.49 | 61.3 | 0.02 | 1.43 | 67.3 | 0.02 |
| HR-Green | 1.22 | 52.7 | 0.03 | 1.59 | 66.0 | 0.03 |
| HR-Infrared | 2.19 | 89.0 | 0.05 | 2.07 | 88.4 | 0.04 |
| HR-Red | 0.73 | 28.2 | 0.01 | 0.82 | 31.5 | 0.01 |
| Avg | 1.23 | 65.8 | 0.02 | 1.08 | 50.8 | 0.02 |

Table 10: **Architecture ablation standard deviations across 5 folds**. Results for both methods are averaged across 5 test folds, and standard deviations can be found in Appendix D. For the classification tasks, higher is better, and evaluation metrics are macro F-1 score (MF1), accuracy (ACC), and the area under the receiver operating characteristic (AUC). For regression tasks lower is better, and we use mean average error (MAE), mean squared error (MSE), and mean average percentage error (MAPE).

| | PaPaGei Arch | | | Proposed Arch | | |
|---|---|---|---|---|---|---|
| Clf (↑) | MF1 | ACC | AUC | MF1 | ACC | AUC |
| Stress | 0.04 | 0.02 | 0.03 | 0.06 | 0.04 | 0.03 |
| Affect | 0.06 | 0.05 | 0.04 | 0.06 | 0.05 | 0.05 |
| Activities | 0.05 | 0.04 | 0.03 | 0.03 | 0.03 | 0.02 |
| Arousal | 0.02 | 0.02 | 0.02 | 0.03 | 0.02 | 0.04 |
| Valence | 0.01 | 0.03 | 0.02 | 0.02 | 0.04 | 0.03 |
| Hypertension | 0.02 | 0.02 | 0.03 | 0.06 | 0.05 | 0.07 |
| Avg | 0.03 | 0.03 | 0.03 | **0.04** | **0.04** | **0.04** |
| Reg (↓) | MAE | MSE | MAPE | MAE | MSE | MAPE |
| HR (DaLiA) | 1.0 | 36.8 | 0.03 | 0.93 | 34.3 | 0.02 |
| Avg-HR | 0.27 | 5.86 | 0.0 | 0.24 | 4.56 | 0.0 |
| Sys-BP | 0.54 | 50.8 | 0.01 | 1.62 | 73.1 | 0.02 |
| Dia-BP | 0.53 | 12.1 | 0.01 | 0.55 | 11.9 | 0.01 |
| Sys-BP (VV) | 1.45 | 98.9 | 0.01 | 2.66 | 169 | 0.02 |
| Dia-BP (VV) | 1.43 | 67.3 | 0.02 | 1.37 | 64.0 | 0.02 |
| HR-Green | 1.59 | 66.0 | 0.03 | 1.36 | 53.7 | 0.03 |
| HR-Infrared | 2.07 | 88.4 | 0.04 | 2.08 | 91.0 | 0.04 |
| HR-Red | 0.82 | 31.5 | 0.01 | 0.41 | 11.9 | 0.01 |
| Avg | 1.08 | 50.8 | 0.02 | 1.25 | 57.1 | 0.02 |

# E    HEART RATE ESTIMATION WITH NEUROKIT ABLATION

Heart rate estimation is often done using automated tools, but in cases where PPG segments are quite noisy, they may fail. In Table 11 we compare the best-performing PaPaGei model in terms of heart rate estimation (PaPaGei-P) with NeuroKit's automatic heart rate estimation, and our proposed model. All models take 10s of PPG segments as input, and for NeuroKit (Makowski et al., 2021) we use `ppg_process`'s `PPG_Rate` output. If no heart rate was detected or not enough peaks were present for NeuroKit, the heart rate was set to 0. Afterwards, we perform the linear probing procedure to account for small linear errors in the NeuroKit model, and to make the procedure as similar to the results reported for PaPaGei-P and our model. Although the example is a little manufactured given that NeuroKit is often used to estimate heart rate for longer segments of PPG data, our experiment provides a one-to-one comparison for real-time 10s window heart rate estimation. Moreover, Table 11 shows that our model outperforms both models, and that PaPaGei outperforms NeuroKit. In some cases, PaPaGei-P and NeuroKit's performances are closely matched, e.g. for HR (DaLiA) and HR-Green.

Table 11: **NeuroKit ablation, NeuroKit heart rate estimation vs. PaPaGei-P and our model**. Results for both methods are averaged across 5 test folds, and standard deviations can be found in Appendix D. For the classification tasks, higher is better, and evaluation metrics are macro F-1 score (MF1), accuracy (ACC), and the area under the receiver operating characteristic (AUC). For regression tasks lower is better, and we use mean average error (MAE), mean squared error (MSE), and mean average percentage error (MAPE).

| | NeuroKit | | | PaPaGei-P | | | Ours | | |
|---|---|---|---|---|---|---|---|---|---|
| Clf (↑) | **MF1** | **ACC** | **AUC** | **MF1** | **ACC** | **AUC** | **MF1** | **ACC** | **AUC** |
| Reg (↓) | **MAE** | **MSE** | **MAPE** | **MAE** | **MSE** | **MAPE** | **MAE** | **MSE** | **MAPE** |
| HR (DaLiA) | 12.9 | 318 | 0.14 | 12.7 | 303 | 0.14 | 7.3 | 139 | 0.08 |
| Avg-HR | 8.67 | 116 | 0.12 | 4.76 | 40.0 | 0.07 | 3.72 | 23.8 | 0.05 |
| HR-Green | 12.1 | 256 | 0.16 | 12.2 | 266 | 0.17 | 7.41 | 149 | 0.1 |
| HR-Infrared | 13.1 | 289 | 0.18 | 12.6 | 273 | 0.17 | 9.81 | 212 | 0.14 |
| HR-Red | 13.1 | 291 | 0.18 | 12.7 | 279 | 0.17 | 10.7 | 233 | 0.15 |
| Avg | 11.9 | 254 | 0.16 | 11.0 | 232 | 0.14 | **7.79** | **151** | **0.1** |

# F EXPANDED BASELINE RESULTS

Table 12: **Additional baseline results**. Results for each method are averaged across 5 test folds, and standard deviations can be found in Appendix D. For the classification tasks, higher is better, and evaluation metrics are macro F-1 score (MF1), accuracy (ACC), and the area under the receiver operating characteristic (AUC). For regression tasks lower is better, and we use mean average error (MAE), mean squared error (MSE), and mean average percentage error (MAPE).

| | BYOL | | | SimCLR | | | PulsePPG | | |
|---|---|---|---|---|---|---|---|---|---|
| Clf (↑) | **MF1** | **ACC** | **AUC** | **MF1** | **ACC** | **AUC** | **MF1** | **ACC** | **AUC** |
| Stress | 0.81 | 0.87 | 0.92 | 0.83 | 0.88 | 0.94 | 0.77 | 0.84 | 0.88 |
| Affect | 0.49 | 0.56 | 0.75 | 0.5 | 0.59 | 0.77 | 0.45 | 0.54 | 0.73 |
| Activities | 0.33 | 0.38 | 0.8 | 0.32 | 0.38 | 0.8 | 0.36 | 0.4 | 0.81 |
| Arousal | 0.51 | 0.57 | 0.55 | 0.52 | 0.56 | 0.56 | 0.5 | 0.55 | 0.54 |
| Valence | 0.52 | 0.61 | 0.58 | 0.53 | 0.61 | 0.58 | 0.54 | 0.6 | 0.57 |
| Hypertension | 0.69 | 0.73 | 0.73 | 0.68 | 0.72 | 0.74 | 0.7 | 0.74 | 0.75 |
| Avg | 0.56 | 0.62 | 0.72 | **0.56** | **0.62** | **0.73** | 0.55 | 0.61 | 0.71 |
| Reg (↓) | **MAE** | **MSE** | **MAPE** | **MAE** | **MSE** | **MAPE** | **MAE** | **MSE** | **MAPE** |
| HR (Dalia) | 9.08 | 176 | 0.1 | 9.49 | 186 | 0.11 | 9.03 | 170 | 0.1 |
| Avg-HR | 3.98 | 29.3 | 0.05 | 4.4 | 33.6 | 0.06 | 4.17 | 30.9 | 0.06 |
| Sys-BP | 13.9 | 314 | 0.11 | 13.6 | 308 | 0.11 | 13.6 | 313 | 0.11 |
| Dia-BP | 8.26 | 110 | 0.12 | 8.31 | 113 | 0.12 | 8.42 | 114 | 0.12 |
| Sys-BP (VV) | 16.8 | 492 | 0.13 | 16.0 | 458 | 0.12 | 15.2 | 400 | 0.11 |
| Dia-BP (VV) | 8.44 | 125 | 0.1 | 8.17 | 122 | 0.1 | 8.09 | 118 | 0.1 |
| HR-Green | 8.67 | 171 | 0.12 | 8.8 | 174 | 0.12 | 9.47 | 188 | 0.13 |
| HR-Infrared | 10.7 | 228 | 0.15 | 10.8 | 230 | 0.15 | 10.8 | 221 | 0.15 |
| HR-Red | 11.4 | 248 | 0.16 | 11.5 | 250 | 0.16 | 12.1 | 261 | 0.17 |
| Avg | 10.1 | 210 | **0.12** | 10.1 | 208 | 0.12 | **10.1** | **202** | 0.12 |

Table 13: **Baselines across subjects linear probing results compared to our model**. Results for each method are averaged across 5 test folds, and standard deviations can be found in Appendix D. For the classification tasks, higher is better, and evaluation metrics are macro F-1 score (MF1), accuracy (ACC), and the area under the receiver operating characteristic (AUC). For regression tasks lower is better, and we use mean average error (MAE), mean squared error (MSE), and mean average percentage error (MAPE).

| | PulsePPG | | | SimCLR | | | Ours | | |
|---|---|---|---|---|---|---|---|---|---|
| Clf (↑) | MF1 | ACC | AUC | MF1 | ACC | AUC | MF1 | ACC | AUC |
| Stress | 0.77 | 0.84 | 0.88 | 0.83 | 0.88 | 0.94 | 0.83 | 0.88 | 0.94 |
| Affect | 0.45 | 0.54 | 0.73 | 0.5 | 0.59 | 0.77 | 0.52 | 0.6 | 0.78 |
| Activities | 0.36 | 0.4 | 0.81 | 0.32 | 0.38 | 0.8 | 0.36 | 0.41 | 0.82 |
| Arousal | 0.5 | 0.55 | 0.54 | 0.52 | 0.56 | 0.56 | 0.53 | 0.57 | 0.57 |
| Valence | 0.54 | 0.6 | 0.57 | 0.53 | 0.61 | 0.58 | 0.53 | 0.62 | 0.58 |
| Hypertension | 0.7 | 0.74 | 0.75 | 0.68 | 0.72 | 0.74 | 0.71 | 0.75 | 0.77 |
| Avg | 0.55 | 0.61 | 0.71 | 0.56 | 0.62 | 0.73 | **0.58** | **0.64** | **0.74** |
| Reg (↓) | MAE | MSE | MAPE | MAE | MSE | MAPE | MAE | MSE | MAPE |
| HR (Dalia) | 9.03 | 170 | 0.1 | 9.49 | 186 | 0.11 | 7.78 | 143 | 0.09 |
| Avg-HR | 4.17 | 30.9 | 0.06 | 4.4 | 33.6 | 0.06 | 3.8 | 26.3 | 0.05 |
| Sys-BP | 13.6 | 313 | 0.11 | 13.6 | 308 | 0.11 | 13.2 | 281 | 0.11 |
| Dia-BP | 8.42 | 114 | 0.12 | 8.31 | 113 | 0.12 | 8.16 | 109 | 0.12 |
| Sys-BP (VV) | 15.2 | 400 | 0.11 | 16.0 | 458 | 0.12 | 15.9 | 451 | 0.12 |
| Dia-BP (VV) | 8.09 | 118 | 0.1 | 8.17 | 122 | 0.1 | 8.04 | 123 | 0.1 |
| HR-Green | 9.47 | 188 | 0.13 | 8.8 | 174 | 0.12 | 7.61 | 149 | 0.1 |
| HR-Infrared | 10.8 | 221 | 0.15 | 10.8 | 230 | 0.15 | 9.82 | 206 | 0.14 |
| HR-Red | 12.1 | 261 | 0.17 | 11.5 | 250 | 0.16 | 10.8 | 231 | 0.15 |
| Avg | 10.1 | 202 | 0.12 | 10.1 | 208 | 0.12 | **9.45** | **191** | **0.11** |

## G MULTIMODAL METRIC ABLATION

Table 14: **ECG vs RESP ablation**. Results for each method are averaged across 5 test folds, and standard deviations can be found in Appendix D. For the classification tasks, higher is better, and evaluation metrics are macro F-1 score (MF1), accuracy (ACC), and the area under the receiver operating characteristic (AUC). For regression tasks lower is better, and we use mean average error (MAE), mean squared error (MSE), and mean average percentage error (MAPE).

| | ECG | | | RESP | | | ECG + RESP | | |
|---|---|---|---|---|---|---|---|---|---|
| Clf (↑) | MF1 | ACC | AUC | MF1 | ACC | AUC | MF1 | ACC | AUC |
| Stress | 0.83 | 0.88 | 0.94 | 0.83 | 0.88 | 0.94 | 0.83 | 0.88 | 0.94 |
| Affect | 0.53 | 0.61 | 0.78 | 0.52 | 0.6 | 0.77 | 0.52 | 0.6 | 0.78 |
| Activities | 0.36 | 0.41 | 0.82 | 0.36 | 0.41 | 0.81 | 0.36 | 0.41 | 0.82 |
| Arousal | 0.52 | 0.54 | 0.53 | 0.53 | 0.57 | 0.56 | 0.53 | 0.57 | 0.57 |
| Valence | 0.54 | 0.61 | 0.57 | 0.54 | 0.61 | 0.58 | 0.53 | 0.62 | 0.58 |
| Hypertension | 0.7 | 0.73 | 0.76 | 0.7 | 0.73 | 0.75 | 0.71 | 0.75 | 0.77 |
| Avg | 0.58 | 0.63 | 0.73 | 0.58 | 0.63 | 0.74 | **0.58** | **0.64** | **0.74** |
| Reg (↓) | MAE | MSE | MAPE | MAE | MSE | MAPE | MAE | MSE | MAPE |
| HR (Dalia) | 6.9 | 123 | 0.08 | 8.97 | 170 | 0.1 | 7.78 | 143 | 0.09 |
| Avg-HR | 3.66 | 23.2 | 0.05 | 4.06 | 28.0 | 0.06 | 3.8 | 26.3 | 0.05 |
| Sys-BP | 13.2 | 289 | 0.11 | 13.4 | 288 | 0.11 | 13.2 | 281 | 0.11 |
| Dia-BP | 8.38 | 113 | 0.12 | 8.3 | 110 | 0.12 | 8.16 | 109 | 0.12 |
| Sys-BP (VV) | 15.9 | 457 | 0.12 | 15.5 | 438 | 0.12 | 15.9 | 451 | 0.12 |
| Dia-BP (VV) | 8.33 | 130 | 0.1 | 8.11 | 116 | 0.1 | 8.04 | 123 | 0.1 |
| HR-Green | 7.17 | 142 | 0.1 | 8.53 | 167 | 0.12 | 7.61 | 149 | 0.1 |
| HR-Infrared | 9.38 | 195 | 0.13 | 10.6 | 222 | 0.15 | 9.82 | 206 | 0.14 |
| HR-Red | 10.4 | 222 | 0.14 | 11.4 | 246 | 0.16 | 10.8 | 231 | 0.15 |
| Avg | **9.25** | **188** | **0.1** | 9.87 | 198 | 0.11 | 9.45 | 191 | 0.11 |

Table 15: **HR vs HR + RESP ablation**. Results for each method are averaged across 5 test folds, and standard deviations can be found in Appendix D. For the classification tasks, higher is better, and evaluation metrics are macro F-1 score (MF1), accuracy (ACC), and the area under the receiver operating characteristic (AUC). For regression tasks lower is better, and we use mean average error (MAE), mean squared error (MSE), and mean average percentage error (MAPE).

| | HR | | | HR + RESP | | | ECG + RESP | | |
|---|---|---|---|---|---|---|---|---|---|
| Clf (↑) | MF1 | ACC | AUC | MF1 | ACC | AUC | MF1 | ACC | AUC |
| Stress | 0.83 | 0.88 | 0.94 | 0.83 | 0.88 | 0.94 | 0.83 | 0.88 | 0.94 |
| Affect | 0.53 | 0.6 | 0.77 | 0.52 | 0.6 | 0.77 | 0.52 | 0.6 | 0.78 |
| Activities | 0.35 | 0.41 | 0.82 | 0.36 | 0.41 | 0.82 | 0.36 | 0.41 | 0.82 |
| Arousal | 0.5 | 0.56 | 0.53 | 0.53 | 0.56 | 0.55 | 0.53 | 0.57 | 0.57 |
| Valence | 0.53 | 0.61 | 0.56 | 0.54 | 0.61 | 0.58 | 0.53 | 0.62 | 0.58 |
| Hypertension | 0.68 | 0.7 | 0.73 | 0.71 | 0.74 | 0.76 | 0.71 | 0.75 | 0.77 |
| Avg | 0.57 | 0.63 | 0.73 | **0.58** | 0.63 | 0.74 | 0.58 | **0.64** | **0.74** |
| Reg (↓) | MAE | MSE | MAPE | MAE | MSE | MAPE | MAE | MSE | MAPE |
| HR (Dalia) | 6.82 | 123 | 0.08 | 8.29 | 153 | 0.09 | 7.78 | 143 | 0.09 |
| Avg-HR | 3.71 | 24.7 | 0.05 | 4.14 | 28.6 | 0.06 | 3.8 | 26.3 | 0.05 |
| Sys-BP | 13.6 | 309 | 0.11 | 13.3 | 289 | 0.11 | 13.2 | 281 | 0.11 |
| Dia-BP | 8.28 | 112 | 0.12 | 8.18 | 109 | 0.12 | 8.16 | 109 | 0.12 |
| Sys-BP (VV) | 16.2 | 453 | 0.12 | 15.9 | 454 | 0.12 | 15.9 | 451 | 0.12 |
| Dia-BP (VV) | 8.1 | 123 | 0.1 | 8.13 | 120 | 0.1 | 8.04 | 123 | 0.1 |
| HR-Green | 7.09 | 139 | 0.1 | 7.99 | 154 | 0.11 | 7.61 | 149 | 0.1 |
| HR-Infrared | 9.33 | 194 | 0.13 | 10.2 | 214 | 0.14 | 9.82 | 206 | 0.14 |
| HR-Red | 10.4 | 223 | 0.14 | 11.2 | 241 | 0.15 | 10.8 | 231 | 0.15 |
| Avg | **9.29** | **189** | **0.1** | 9.7 | 196 | 0.11 | 9.45 | 191 | 0.11 |

# H  CONTRASTIVE LEARNING ABLATION

Table 16: **Our model with L2 vs the contrastive loss**. Results for each method are averaged across 5 test folds, and standard deviations can be found in Appendix D. For the classification tasks, higher is better, and evaluation metrics are macro F-1 score (MF1), accuracy (ACC), and the area under the receiver operating characteristic (AUC). For regression tasks lower is better, and we use mean average error (MAE), mean squared error (MSE), and mean average percentage error (MAPE).

| | Ours (L2 loss) | | | Ours (new) | | |
|---|---|---|---|---|---|---|
| Clf (↑) | MF1 | ACC | AUC | MF1 | ACC | AUC |
| Stress | 0.81 | 0.86 | 0.92 | 0.83 | 0.88 | 0.94 |
| Affect | 0.5 | 0.58 | 0.76 | 0.52 | 0.6 | 0.78 |
| Activities | 0.31 | 0.37 | 0.79 | 0.36 | 0.41 | 0.82 |
| Arousal | 0.51 | 0.56 | 0.54 | 0.53 | 0.57 | 0.57 |
| Valence | 0.53 | 0.61 | 0.59 | 0.53 | 0.62 | 0.58 |
| Hypertension | 0.69 | 0.73 | 0.73 | 0.71 | 0.75 | 0.77 |
| Avg | 0.56 | 0.62 | 0.72 | **0.58** | **0.64** | **0.74** |
| Reg (↓) | MAE | MSE | MAPE | MAE | MSE | MAPE |
| HR (Dalia) | 8.39 | 163 | 0.09 | 7.78 | 143 | 0.09 |
| Avg-HR | 3.84 | 26.1 | 0.05 | 3.8 | 26.3 | 0.05 |
| Sys-BP | 14.1 | 330 | 0.11 | 13.2 | 281 | 0.11 |
| Dia-BP | 8.64 | 121 | 0.12 | 8.16 | 109 | 0.12 |
| Sys-BP (VV) | 15.6 | 433 | 0.12 | 15.9 | 451 | 0.12 |
| Dia-BP (VV) | 8.37 | 126 | 0.1 | 8.04 | 123 | 0.1 |
| HR-Green | 8.52 | 173 | 0.12 | 7.61 | 149 | 0.1 |
| HR-Infrared | 10.8 | 237 | 0.15 | 9.82 | 206 | 0.14 |
| HR-Red | 11.6 | 255 | 0.16 | 10.8 | 231 | 0.15 |
| Avg | 9.98 | 207 | 0.11 | **9.45** | **191** | **0.11** |

# I PROJECTOR ABLATION

Given the success of the BYOL and SimCLR baselines, we decided to pre-train a new version of our model that also has a projector that we calculate the Rank-N-Contrast loss on, instead of on the embeddings directly. We found gains in both the classification and regression tasks, and use the updated model for all experiments.

Table 17: **New model (with projector) vs original model across subjects linear probing results**. Results for each method are averaged across 5 test folds, and standard deviations can be found in Appendix D. For the classification tasks, higher is better, and evaluation metrics are macro F-1 score (MF1), accuracy (ACC), and the area under the receiver operating characteristic (AUC). For regression tasks lower is better, and we use mean average error (MAE), mean squared error (MSE), and mean average percentage error (MAPE).

| | Ours | | | Ours (new) | | |
|---|---|---|---|---|---|---|
| Clf (↑) | **MF1** | **ACC** | **AUC** | **MF1** | **ACC** | **AUC** |
| Stress | 0.82 | 0.87 | 0.93 | 0.83 | 0.88 | 0.94 |
| Affect | 0.53 | 0.62 | 0.78 | 0.52 | 0.6 | 0.78 |
| Activities | 0.35 | 0.41 | 0.81 | 0.36 | 0.41 | 0.82 |
| Arousal | 0.53 | 0.56 | 0.55 | 0.53 | 0.57 | 0.57 |
| Valence | 0.55 | 0.62 | 0.59 | 0.53 | 0.62 | 0.58 |
| Hypertension | 0.68 | 0.72 | 0.75 | 0.71 | 0.75 | 0.77 |
| Avg | 0.58 | 0.63 | 0.74 | **0.58** | **0.64** | **0.74** |
| Reg (↓) | **MAE** | **MSE** | **MAPE** | **MAE** | **MSE** | **MAPE** |
| HR (Dalia) | 7.2 | 134 | 0.08 | 7.78 | 143 | 0.09 |
| Avg-HR | 3.74 | 24.9 | 0.05 | 3.8 | 26.3 | 0.05 |
| Sys-BP | 13.5 | 299 | 0.11 | 13.2 | 281 | 0.11 |
| Dia-BP | 8.38 | 115 | 0.12 | 8.16 | 109 | 0.12 |
| Sys-BP (VV) | 16.6 | 473 | 0.12 | 15.9 | 451 | 0.12 |
| Dia-BP (VV) | 9.0 | 145 | 0.11 | 8.04 | 123 | 0.1 |
| HR-Green | 7.28 | 144 | 0.1 | 7.61 | 149 | 0.1 |
| HR-Infrared | 9.57 | 201 | 0.13 | 9.82 | 206 | 0.14 |
| HR-Red | 10.6 | 227 | 0.15 | 10.8 | 231 | 0.15 |
| Avg | 9.54 | 196 | 0.11 | **9.45** | **191** | **0.11** |

