# OpenReview forum: "A robust PPG foundation model using multimodal physiological supervision"
_ICLR.cc/2026/Conference — Submitted to ICLR 2026_

### Official Review · Reviewer_PDvh · 2025-10-30

**Soundness:** 2
**Presentation:** 3
**Contribution:** 2
**Rating:** 4
**Confidence:** 4

**Summary:**

The paper proposes a robust PPG foundation model that leverages ECG and respiratory signals to guide sample selection during contrastive learning. The paper emphasizes that by integrating complementary biosignal modalities, the proposed approach effectively mitigates the limitations of unimodal, morphology-based contrastive targets, resulting in substantially improved robustness, generalization, and downstream task performance.

**Strengths:**

- The paper’s focus on multi-modal supervision is well-motivated, particularly given that health is inherently multi-modal while most existing foundation models remain unimodal.
- In general, the ablation and case studies offer meaningful insights and contribute to understanding the model’s behavior and robustness.
- The paper is clearly written, well-organized, and easy to follow.

**Weaknesses:**

**Method:** The proposed approach employs five key physiological metrics from ECG and RESP: HR, RMSSD, RR, RA, and RV to guide multi-modal supervision during pre-training, enabling the model to learn corresponding representations in the latent space. However, many of the downstream tasks evaluated (e.g., HR, BP, and activity recognition) are directly related to these same input measures. This raises concerns about potential task overlap and limited generalization. How can this design choice be justified? If the input features and downstream tasks are closely aligned, it becomes unclear whether the learned representations truly generalize beyond the pre-training objectives. Perhaps, consider evaluations on tasks that are novel and not previously explored.

**Experimental Design**:
There are two major concerns with the experimental design: **(1) the choice of baseline** and **(2) the use of derived metrics for training**.

First, the experiments rely heavily on PaPaGei as the sole baseline. While PaPaGei is a reasonable point of comparison, it cannot be the only one. The key issue is that PaPaGei is trained exclusively on PPG signals, whereas the proposed model leverages multi-modal supervision, including ECG and respiratory signals. Comparing a unimodal foundation model with a multi-modal supervised one introduces an inherent imbalance and may not provide a fair assessment of performance gains.

Second, the rationale for using derived physiological metrics (HR, RMSSD, RR, RA, and RVT) during pre-training needs stronger justification, especially since these metrics are closely correlated with several downstream tasks. It remains unclear whether their inclusion in pre-training offers benefits beyond what could be achieved by incorporating them at the linear probing stage alongside the learned embeddings. More fundamentally, how would a simple baseline model trained directly on these five derived features perform on the same downstream tasks? Addressing this question would help clarify the true contribution of the proposed approach.

**Minor:**
Figure 3 — The comparison of UMAP plots for heart rate may not be meaningful, as the proposed approach uses heart rate as part of its pre-training objectives, whereas the baseline models do not. Consequently, it is expected that the proposed model exhibits a clearer gradient structure in the latent space, which limits the interpretive value of this comparison.

**Ablation Study:** It would be valuable to analyze the individual contribution of each computed metric derived from the co-recorded signals. Specifically, examining the effect of using HR, RMSSD, RR, RA, and RVT during pre-training, either by incorporating one metric at a time or by comparing groups of metrics (e.g., ECG-based vs. respiratory-based). This could provide deeper insights into which modalities or features most influence model performance.

**Open-Source**: The models and code are not publicly released, even though the proposed approach is trained and evaluated on open-source datasets. This limits the reproducibility of the work and weakens its overall contribution, particularly given that other open-source PPG foundation models already exist [1, 2].

Overall, the main contribution of this work appears to be the inclusion of additional biosignal modalities during contrastive pre-training, which improves the performance of a PPG foundation model. While this idea is interesting, the contribution is not sufficiently significant, as prior studies have already explored unimodal versus multimodal representations in similar contexts [3, 4]. Importantly, given the limitations in the experimental design, methodological justification, and lack of open-source release discussed above, I lean toward a weak reject recommendation.

[1] Pillai, A., Spathis, D., Kawsar, F., & Malekzadeh, M. (2024). Papagei: Open foundation models for optical physiological signals. _arXiv preprint arXiv:2410.20542_.

[2] Saha, M., Xu, M. A., Mao, W., Neupane, S., Rehg, J. M., & Kumar, S. (2025). Pulse-ppg: An open-source field-trained ppg foundation model for wearable applications across lab and field settings. _Proceedings of the ACM on Interactive, Mobile, Wearable and Ubiquitous Technologies_, _9_(3), 1-35.

[3] Zhou, Y., Khasentino, J., Yun, T., Biradar, M. I., Shreibati, J., Lai, D., ... & Hormozdiari, F. (2025). Applying multimodal AI to physiological waveforms improves genetic prediction of cardiovascular traits. _The American Journal of Human Genetics_.

[4] Ezzameli, K., & Mahersia, H. (2023). Emotion recognition from unimodal to multimodal analysis: A review. _Information Fusion_, _99_, 101847.

**Questions:**

- Are there other ECG and RESP metrics that can be used during contrastive pre-training?

---

> ### Author Response · Authors · 2025-11-25
> **Reviewer PDvh response 1**
>
> We appreciate the reviewer’s insightful feedback, particularly their emphasis on incorporating additional baselines and their discussion of how pre-training metrics relate to downstream labels.
>
> ## [PDvh-W1]: Multimodal metric overlap with downstream labels
> We understand the reviewer’s concern about correlations between downstream labels and multimodal signals, but we believe this correlation is unavoidable, and we have not found any downstream tasks that do not have such a correlation. This is mainly because PPG was developed to provide a cheap and non-invasive measure of blood volume changes, which is biophysically related to the heart and a person’s respiration.
>
> From a foundation model perspective, they are pre-trained on large-scale data such that they form a base for a wide variety of tasks. ‘Tasks’ can mean many things, and differ based on the field and type of signal under consideration. In case of PPG, we believe generalization and tasks mostly vary along three main axes:
> Prediction labels: The exact downstream task that is evaluated, i.e. stress, valence, heart rate, blood pressure, etc.
> Data distributions: PPG signals from either different locations on the body, from different wavelengths, or from different types of recording settings (clinical vs lab vs field)
> People: Generalization to unseen demographics/groups of people
>
> Specifically, one of the main conclusions in the PulsePPG paper [1] is that “This suggests that exposure to real-world variability enables the model to learn fine-grained representations, making it more adaptable across tasks.”. This is an important result indicating that pre-training a model with clinical data does not allow it to easily generalize to real-world data. Given that large-scale field-like datasets are generally closed-source, our results are highly encouraging and suggest that PPG foundation models can generalize from clinical to real-world datasets as long as they make use of the inherent multimodality of many clinical datasets. Thus, label generalization is not the only type of desirable generalization for PPG foundation models. Moreover, any downstream dataset/task we could find that involves PPG will in some way be related to ECG/RESP or other potential metrics as well. This is largely because, as you put it **[PDvh-S1]** “... health is inherently multi-modal …”, and most PPG datasets obtain measures that directly (heart rate, blood pressure) or indirectly (stress, valence) relate to a person’s heart or respiration. Utilizing high-quality multimodal signals from clinical datasets that enrich embeddings with more heart and respiration-related information is thus a natural improvement. The improvement may seem obvious, but focusing only on this type of generalization diminishes the importance of the latter two types of generalization, which are especially important given the highly unique data we use to pre-train the model. Namely, we show that our model, although pre-trained on clinical data, generalizes to completely different data distributions, WildPPG and Dalia, where one of the main goals is to accurately infer heart rate using wearables as opposed to requiring an ECG patch, which is extremely impractical. Lastly, both through results on data obtained from young people (EEGVR and WESAD) and with our within-subject evaluation framework we show that our model generalizes from a specific group of people (people in the ICU) to very different distributions of people. So even though the metrics are related to downstream tasks, some of our main contributions relate to the other two types of generalization as well.
>
> Lastly, we specifically added the EEGVR dataset because it focuses more on affective computing in a natural setting, and measures valence and arousal. Additionally, the WildPPG dataset was specifically designed as a tough-to-generalize type of PPG data because it records variations in PPG location (chest, wrist, etc.), PPG measuring frequency, and activity during which PPG is recorded.

---

> > ### Author Response · Authors · 2025-11-25
> > **Reviewer PDvh response 2**
> >
> > ## [PDvh-W2]: Additional baselines and unimodal vs multimodal foundation models
> > We appreciate the reviewer’s comment. We would like to clarify that during inference, our model uses only PPG signals, despite being pretrained with multi-modal signals. The ECG and respiratory signals are leveraged solely during pre-training to guide the model and obtain more robust PPG representations. Therefore, at test time, the model does not have access to these multimodal signals, and the comparison with PaPaGei (which is trained and tested on PPG alone) remains meaningful. The performance gains are thus due to improved PPG representation learning that is more robust to data shifts and noise, rather than access to additional modalities during inference.
> > To address the limited number of baselines, we have added evaluations for BYOL, SimCLR, and PulsePPG to Appendix F. Our results show that all baselines on average underperform our proposed model for both classification and regression tasks. The only two tasks that baselines outperform our model at are the macro F-1 score for valence classification (PulsePPG F-1: 0.54, ACC: 0.6, AUC: 0.57 vs Ours F-1: 0.53, ACC: 0.62, AUC: 0.58) and Systolic blood pressure regression for the VitalVideos dataset (PulsePPG MAE: 15.2 vs Ours MAE: 15.9). An important sidenote to these results is that we use PulsePPG’s pre-trained weights, which are obtained after pre-training on a large-scale closed-source field-like dataset. In contrast, we use a relatively small publicly available clinical dataset.
> >
> > ## [PDvh-W3]: Multimodal metric overlap and baseline model
> > We explain why we believe multimodal metric overlap is not fundamentally an issue in our response to **[PDvh-W1]**.
> >
> > Regarding a baseline model that is trained directly on the five metrics, we have added this model as an ablation study to Appendix H. We find that our contrastive learning objective outperforms this regression-based model on all but one task. We want to thank the reviewer for this suggestion because we believe this ablation further signifies that our current approach has the highest impact, but also because our current approach easily allows for the inclusion of additional metrics, whereas a baseline model trained directly to predict specific metrics does not.
> >
> > ## [PDvh-W4] Relevance of UMAP Plots
> > We explain why we believe multimodal metric overlap is not fundamentally an issue in our response to **[PDvh-W1]**.
> >
> > Regarding the specific UMAP plots, we specifically chose to visualize the Dalia dataset because it is a field-like dataset, which means the gradient structure also visualizes how well each PPG foundation model is able to generalize to real-life data. To address this comment, we will clarify this point in a revised manuscript.
> >
> > ## [PDvh-W5]: Individual contributions of each computed metric
> > We appreciate the reviewer’s comment, and have added an ablation study to Appendix G to address it. Our ablation study finds that pre-training only with ECG performs best on average at regression tasks, and ECG + RESP performs best on average for classification tasks. We note that the ECG + RESP model outperforms all baselines across the largest number of tasks, so we conclude that it is the most versatile model. Moreover, we find that both ECG-only and RESP-only pre-training improves performance, even for downstream tasks that are not related to the multimodal signal. Specifically, the RESP-only pre-training outperforms all baselines on all but one heart rate regression tasks, even though we do not provide the model with explicit heart rate information during pre-training, and only use RESP-derived metrics. Overall we thus find that ECG and ECG + RESP are the best models, but that any pre-train metric improves performance, **even on downstream tasks that are not directly related to the metrics**.

---

> ### Author Response · Authors · 2025-11-25
> **Reviewer PDvh response 3**
>
> ## [PDvh-W6]: Limited availability of open-source code and weights
> We thank the reviewer for this comment, and we will add to our reproducibility statement that although the model/weights are under an NDA, we are able to help people with implementations if they reach out to us.
>
> ## [PDvh-Q1]: Are there other ECG and RESP metrics that can be used during contrastive pre-training?
> For the RESP metrics, we have tried to include as many metrics as are easily computable, and for the ECG metrics we specifically chose metrics based on a large-scale meta-study [2]. This meta study indicates that HR and RMSSD are the main robust measures that can be estimated from ECG within a 10s window. We will add in our future work section that MIMIC-3 contains other multimodal signals as well, such as arterial blood pressure (ABP), SpO2, and systolic and diastolic blood pressure. The latter three are only included periodically, and to ensure that including all multimodal signals does not reduce the size of the pre-training dataset, future work should explore ways to deal with missing values for either the continuous multimodal signals or periodic measures. We will highlight this future direction in our revised manuscript.
>
> References: \
> [1] Saha, M., Xu, M. A., Mao, W., Neupane, S., Rehg, J. M., & Kumar, S. (2025). Pulse-ppg: An open-source field-trained ppg foundation model for wearable applications across lab and field settings. Proceedings of the ACM on Interactive, Mobile, Wearable and Ubiquitous Technologies, 9(3), 1-35. \
> [2] Schroeder, E. B., Whitsel, E. A., Evans, G. W., Prineas, R. J., Chambless, L. E., & Heiss, G. (2004). Repeatability of heart rate variability measures. Journal of electrocardiology, 37(3), 163-172.

---

### Official Review · Reviewer_zKep · 2025-10-30

**Soundness:** 2
**Presentation:** 2
**Contribution:** 2
**Rating:** 4
**Confidence:** 4

**Summary:**

This paper introduces a PPG foundation model pretrained on multimodal ICU data, where synchronized ECG and respiratory signals guide contrastive learning to derive robust and generalizable PPG representations. Compared with prior single-modality approaches (e.g., PaPaGei), the model leverages noise and signal variability more effectively, achieving substantial performance gains on 14 out of 15 downstream tasks across six unseen datasets.

**Strengths:**

1. This study leverages ECG and respiratory signals to enhance PPG representation learning, demonstrating a solid understanding of the physiological mechanisms underlying PPG.
2. It evaluates both cross-subject and within-subject settings, providing a comprehensive view of the model’s generalization performance.
3. The proposed model achieves significant performance improvements over PaPaGei across multiple downstream tasks.

**Weaknesses:**

Key Concerns:
1. The study compares the proposed model only with PaPaGei, which limits the comprehensiveness of the evaluation. It is recommended to include additional self-supervised or pretrained baselines (e.g., SimCLR, PulsePPG) to strengthen the experimental validity.
2. The paper claims that the inherent noise and variability in ICU data can be leveraged to improve model robustness. However, the proposed pretraining approach relies on five contrastive objectives computed from synchronized ECG and respiratory signals. It remains unclear how the authors ensure that these auxiliary signals remain reliable when the PPG signal quality deteriorates (e.g., due to patient motion). As a result, this claim may be somewhat overstated, since the method appears to focus more on multimodal assistance for PPG representation learning rather than on directly addressing signal quality issues.
3. Although the proposed model performs well on downstream tasks, concerns remain regarding its cross-dataset generalization. The model is pretrained solely on the MIMIC dataset, while larger and more diverse publicly available datasets such as MESA or VitalDB could have been incorporated to build a more comprehensive pretraining corpus. Although the authors mention that sleep or anesthesia data may contain relatively stationary signals, incorporating more heterogeneous datasets could capture a wider range of physiological patterns and improve the model’s robustness and generalization.
4. The method employs only derived features from ECG and respiratory signals instead of the raw multimodal inputs, which may constrain the model’s ability to capture complex temporal dependencies.

Minor Concerns:
1. Although the number of subjects used is one-third of that in PaPaGei, the total number of data segments is comparable, thus the claim of higher data efficiency is not entirely justified.
2. The method employs only derived features from ECG and respiratory signals instead of the raw multimodal inputs, which may constrain the model’s ability to capture complex temporal dependencies.
3. Table 1 does not report the number of subjects and total samples for each downstream dataset, which makes it somewhat difficult to fully assess data balance and generalization stability.

**Questions:**

1. Could the authors clarify whether the auxiliary signals (ECG and respiration) remain reliable for contrastive supervision when the PPG signal quality is low, for instance due to patient motion?
2. Would it be possible to include additional self-supervised baselines such as SimCLR, BYOL, or PulsePPG for a more comprehensive comparison?
3. Could the authors elaborate on the rationale for using derived metrics instead of full multimodal inputs, and whether any experiments were conducted to validate this design choice?
4. It would be helpful to provide statistics on the number of subjects and total samples for each downstream dataset, to better contextualize task scale and model performance.
5. Incorporating downstream tasks related to cardiac arrhythmias (e.g., atrial fibrillation) could further demonstrate the model’s ability to handle abnormal cardiac patterns.

---

> ### Author Response · Authors · 2025-11-25
> **Reviewer zKep response 1**
>
> We thank the reviewer for their helpful comments, particularly regarding additional baselines and the discussion of generalization and pre-training dataset sizes.
>
> ## [zKep-W1,Q2]: Additional baselines
> We agree with the reviewer and we have added additional baseline results (BYOL, SimCLR, and PulsePPG) to Appendix F. Our results show that all baselines on average underperform our proposed model for both classification and regression tasks. The only two tasks that baselines outperform our model at are the macro F-1 score for valence classification (PulsePPG F-1: 0.54, ACC: 0.6, AUC: 0.57 vs Ours F-1: 0.53, ACC: 0.62, AUC: 0.58) and Systolic blood pressure regression for the VitalVideos dataset (PulsePPG MAE: 15.2 vs Ours MAE: 15.9). An important sidenote to these results is that we use PulsePPG’s pre-trained weights, which are obtained after pre-training on a large-scale closed-source field-like dataset. In contrast, we use a relatively small publicly available clinical dataset.
>
> ## [zKep-W2,Q1]: Multimodal reliability during PPG degradation
> We agree with the reviewer that it is important to ensure that the physiological parameters we use during contrastive learning are robust. In Appendix A we discuss the steps we take to ensure that although the metrics are estimated in 10s windows they are robust. Namely, we ensure each metric is within normal physiological ranges, and is filtered together with the other 10s windows in each session. This ensures that any outliers are picked up on and filtered + kept within a normal physiological range. Moreover, we believe by the nature of how they are acquired compared to the PPG sensor, even without the additional pre-processing we do, the multimodal signals we use are more robust. Specifically, the ECG signal is acquired at the subject’s chest, whereas the PPG signal is acquired from the subject’s finger. A smaller amount of movement in the chest area compared to a finger, and the closeness to the heart ensure that the signals are inherently more robust than the PPG signal, which we believe still improves representation learning. By choosing robust multimodal signals that are uniquely available in clinical datasets and choosing metrics that we can clip and easily filter across the full recording session, we are able to retain all PPG segments with high movement or other types of noise. This induces a greater robustness to natural variations in noise that allow our model to generalize well to field-like datasets and non-clinical populations. We also perform an additional ablation study that is included in Appendix G. Our ablation study finds that pre-training only with ECG performs best on average at regression tasks, and ECG + RESP performs best on average for classification tasks. We note that the ECG + RESP model outperforms all baselines across the largest number of tasks, so we conclude that it is the most versatile model. Moreover, we find that both ECG-only and RESP-only pre-training obtain high performance models. Additionally, the ECG results (HR + RMSSD) are better than the HR-only or HR-RESP results, so we conclude that even the potentially least robust metric (RMSSD) is robust enough to improve pre-training for PPG foundation models.
>
> ## [zKep-W3,Q3]: Raw multimodal inputs vs physiological metrics
> The main reason we decided to use physiological metrics vs raw multimodal inputs has to do with the filtering choices we lay out in Appendix A. Namely, to ensure the physiological metrics are robust, we need to pre-process the multimodal signals, especially in cases where there is significant motion, for example, that affects both the multimodal signals as well as the PPG signal. By selecting specific physiological parameters, we can use their known biological ranges, and filter the individual values using the rest of the session, i.e. a change in heart rate within 10s is physiologically limited. On the other hand, this type of filtering is not trivial for raw signals, i.e. it is unclear how we can use the rest of the session to interpolate or filter individual 10s segments of the ECG/RESP signal. Our design choice thus goes back to **[zKep-W2,Q1]**, that physiological ranges allow us to pre-process the multimodal signals in such a way that they are as robust as possible. We do believe using raw multimodal inputs is an interesting future direction, and we will highlight it in a revised version of our manuscript.

---

> > ### Author Response · Authors · 2025-11-25
> > **Reviewer zKep response 2**
> >
> > ## [zKep-W4]: Cross-dataset generalization
> > We definitely agree with the reviewer that it can be beneficial to increase the size of the dataset, and adding pre-training datasets would, as the reviewer points out, increase the heterogeneity of the data, and a wider range of physiological patterns. In light of the reviewer’s assessment however, we believe our downstream generalization results are even stronger and truly highlight how multimodal signals can increase data efficiency and cross-dataset generalization. Specifically, even though we only use a single dataset during pre-training, we find large improvements on completely unseen downstream tasks that are both different in the population (clinical vs non-clinical) and the type of data acquisition (clinical vs lab vs field data, i.e. Dalia and WildPPG). Performance improvements thus truly arise from our use of multimodal signals during pre-training as opposed to different underlying datasets, further solidifying our claim. To further improve performance and as the reviewer accurately puts it ‘... capture a wider range of physiological patterns…’, we will emphasize the potential of further increasing the pre-training dataset in the future work section.
> >
> > ## [zKep-W5]: Number of pre-training data segments vs pre-training subjects
> > We understand that the reviewer brings this point up, and we agree that in terms of number of segments the amount of pre-training data we use is the same. From a data generation process perspective, however, the reason we use the same number of segments with 3x fewer subjects and only one out of three datasets, is because we can utilize essentially all PPG segments that are recorded. In terms of the data generation process, it takes much more time to generate three datasets than one, and a model that can improve performance with only one of three datasets because it can utilize all PPG segments during pretraining, is much more data efficient. To the reviewer’s own point **[zKep-W4]**, we show that with a much smaller range of physiological patterns, and by utilizing the multimodal signals that are available in the MIMIC dataset, we can achieve better performance. In our revised manuscript we will change how we emphasize pre-training data efficiency to avoid a potentially confusing claim, and explain our reasoning behind the data efficiency claim more. We thank the reviewer for this suggestion because we see now that the way we phrased it in the submitted manuscript, the comparison is confusing.
> >
> > ## [zKep-W6,Q4]: Number of downstream subjects and samples reporting
> > We thank the reviewer for pointing this out, and we have added the total number of subjects and samples into Table 1 in the revised manuscript.
> >
> > ## [zKep-Q5]: Incorporating downstream tasks related to cardiac arrhythmias (e.g., atrial fibrillation) could further demonstrate the model’s ability to handle abnormal cardiac patterns.
> > We appreciate the reviewer’s question, and we agree that increasing the number of downstream tasks would further help evaluate each model. However, given the limited time in the rebuttal we will recommend additional downstream tasks in the discussion of our revised manuscript.

---

### Official Review · Reviewer_ye6c · 2025-11-07

**Soundness:** 2
**Presentation:** 2
**Contribution:** 2
**Rating:** 4
**Confidence:** 4

**Summary:**

The paper proposes a PPG foundation model pretrained on noisy ICU PPG by using ECG and respiration only during pretraining to compute HR, RMSSD, breathing rate, breathing amplitude, and RVT. A rank-n-contrast loss then pulls PPG embeddings closer when the targets are similar, aiming to learn noise-robust representations.

**Strengths:**

Clear, physiologically grounded supervision: Using ECG/RESP to form targets avoids brittle PPG morphology extraction, yet preserves unimodal inference. The target set (HR, RMSSD, RR, RA, RVT) is plausible for 10-s windows and filtered for physiological ranges.

**Weaknesses:**

- The contribution of the method is trivial and uses the common infoNCE loss. The authors only consider the physiological parameters to decide the positive and negative pairs.
- The robustness of HR/RMSSD/RR/RA/RVT estimation on 10-s windows (detectors, failure handling, thresholds) is crucial; more explicit error rates/quality filters would strengthen claims about the stability of the metric space.
- The final backbone checkpoint is chosen by VitalVideos systolic BP probe performance for practicality. This could inadvertently bias toward that dataset/task; a small sensitivity analysis (random/earliest/best-avg across a subset) would help.
- The final backbone is ~28.8M params vs PaPaGei’s ~5–5.7M. The architecture ablation shows gains even when the architecture differs, but fully disentangling capacity from supervision remains tricky without equal-capacity baselines for all comparisons.
- No fine-tuning heads or end-to-end adaptation are reported; it’s unclear how the model behaves under modest supervised finetuning, which is typical in practice.

**Questions:**

Please see the weakness part.

---

> ### Author Response · Authors · 2025-11-25
> **Reviewer ye6c response 1**
>
> We appreciate the reviewer’s insightful feedback and their encouragement to strengthen our evaluation and generalization arguments and experiments.
>
> ## [ye6c-W1]: Trivial contribution
> We would like to clarify that the main contribution of our work is not the specific choice of loss function, but the demonstration that incorporating multi-modal physiological supervision substantially improves representation learning and out-of-distribution data generalization for PPG signals. While we use Rank-N-Contrast (an InfoNCE-based formulation) as the implementation, this is merely one practical choice; alternative contrastive objectives or training strategies could be used without affecting our core claim.
> To the best of our knowledge, this is the first work in the PPG foundation-model literature that leverages multi-modal physiological parameters to define positive and negative relationships for contrastive learning. The methodological novelty lies in introducing these structured physiological signals as pre-training cues and the resulting model to generalize well to field-like and non-clinical datasets, not in the loss function itself. To further emphasize this point, we have added an ablation study comparing the use of a supervised loss function on the physiological parameters directly and our contrastive pre-training strategy in Appendix H. Our results show improvements both in terms of classification and regression performance across all but one task, which indicates that just training on the metrics directly is not as effective as using our contrastive learning approach for a foundation model. We believe this largely stems from overfitting, and it also aligns with the results in the original Rank-N-Contrast paper [1].
>
> ## [ye6c-W2]: Metric estimation robustness in 10s windows
> We agree with the reviewer that it is important to ensure that the physiological parameters we use during contrastive learning are robust. In Appendix A we discuss the steps we take to ensure that although the metrics are estimated in 10s windows they are robust. Namely, we ensure each metric is within normal physiological ranges, and is filtered together with the other 10s windows in each session. This ensures that any outliers are picked up on and filtered + kept within a normal physiological range. Moreover, we believe by the nature of how they are acquired compared to the PPG sensor, even without the additional pre-processing we do, the multimodal signals we use are more robust. Specifically, the ECG signal is acquired at the subject’s chest, whereas the PPG signal is acquired from the subject’s finger. A smaller amount of movement in the chest area compared to a finger, and the closeness to the heart ensure that the signals are inherently more robust than the PPG signal, which we believe still improves representation learning. Lastly, we specifically chose the ECG-derived metrics based on a large-scale meta-study [2] we cite that indicates both HR and RMSSD are the most robust measures in 10s windows. There is also literature that indicates it is plausible to get accurate respiratory metrics from 10s windows [3], but to ensure that we see improved performance when including the respiratory metrics, we perform an additional ablation study that is included in Appendix G. Our ablation study finds that pre-training only with ECG performs best on average at regression tasks, and ECG + RESP performs best on average for classification tasks. We note that the ECG + RESP model outperforms all baselines across the largest number of tasks, so we conclude that it is the most versatile model. Moreover, we find that both ECG-only and RESP-only pre-training obtain high performance models. Additionally, the ECG results (HR + RMSSD) are better than the HR-only or HR-RESP results, so we conclude that even the potentially least robust metric (RMSSD) is robust enough to improve pre-training for PPG foundation models.

---

> > ### Author Response · Authors · 2025-11-25
> > **Reviewer ye6c response 2**
> >
> > ## [ye6c-W3]: Checkpoint selection bias
> > We acknowledge the reviewer’s concern about potential bias from selecting the model checkpoint using the VitalVideos systolic BP probe. In our revision we expand the number of tasks we include to evaluate checkpoints. Specifically, we include Stress, Affect, Activities, Arousal, Valence, Hypertension, HR (Dalia), Avg-HR, Sys-BP, Dia-BP, Sys-BP (VV), and Dia-BP (VV). This means that only WildPPG is not used to evaluate checkpoints, since we decided to keep at least one dataset completely out of distribution for evaluation. Note that we evaluate checkpoints only on the training/validation sets, so the test set for each checkpoint is unseen. Given that the performance metrics each have different magnitudes and different signs, we decided to z-score (with the median and mean absolute deviation to ensure z-scores were robust) each metric across checkpoints. Then, we took the median z-score across all metrics to obtain a median z-score for each checkpoint, and used this median z-score to select the best checkpoint. Performing this more robust checkpoint selection improved the average downstream classification performance by 1.7% for our model, and we have adopted it for all of the results we have updated in our revised manuscript.
> >
> > ## [ye6c-W4]: PaPaGei scaling ablation
> > We understand the reviewer’s concern, but the reason we did not include additional scaling results for PaPaGei to match parameters is because in their paper (Figure 6c) they show that increasing parameters from 5M to 35 and 139M did not improve performance, and in most tasks decreased performance. If the reviewer believes this experiment is important in the decision to accept this paper, we would be happy to try and do the experiment within the rest of the rebuttal period.
> >
> > ## [ye6c-W5]: Fine-tuning results
> > We do agree with the reviewer that fine-tuning may further improve downstream performance, but the focus of our paper is building a robust PPG foundation model and its *zero-shot* application on downstream tasks to show its high generalization ability. Specifically, we demonstrate that multimodal signals improve PPG pre-training in a way that allows it to generalize to field-like datasets (Dalia and WildPPG), and a variety of non-clinical populations. We believe that our linear probing results provide enough validity for this claim, and we followed PaPaGei in only reporting linear probing results. We do believe fine-tuning is an important future work, and we will add this into our discussion section in a revised version of our manuscript.
> >
> > References: \
> > [1] Zha, K., Cao, P., Son, J., Yang, Y., & Katabi, D. (2023). Rank-n-contrast: learning continuous representations for regression. Advances in Neural Information Processing Systems, 36, 17882-17903. \
> > [2] Schroeder, E. B., Whitsel, E. A., Evans, G. W., Prineas, R. J., Chambless, L. E., & Heiss, G. (2004). Repeatability of heart rate variability measures. Journal of electrocardiology, 37(3), 163-172. \
> > [3] Karlen, W., Gan, H., Chiu, M., Dunsmuir, D., Zhou, G., Dumont, G. A., & Ansermino, J. M. (2014). Improving the accuracy and efficiency of respiratory rate measurements in children using mobile devices. PloS one, 9(6), e99266.

---

### Author Response · Authors · 2025-11-25
**General response**

We want to thank all three reviewers for taking the time to review our work, and for their thoughtful feedback. Moreover, we are happy that the reviewers recognize some key strengths in our paper:

- **ye6c**: “Clear, physiologically grounded supervision: Using ECG/RESP to form targets avoids brittle PPG morphology extraction, yet preserves unimodal inference.”
- **zKep**: “The proposed model achieves significant performance improvements over PaPaGei across multiple downstream tasks.”
- **PDvh**: “The paper’s focus on multi-modal supervision is well-motivated, particularly given that health is inherently multi-modal while most existing foundation models remain unimodal.”

The reviewers’ suggestions have allowed us to significantly improve the updated manuscript, and further improve our model’s results. We will address each individual reviewer’s comments in a thread below their review, but we have decided to address some of the common themes in this general response. All additional experiments have been added to Appendices F-I, and Tables 1 (datasets), 4 (unimodal vs multimodal), and 5 (architectural ablation) have been updated in the main text. We will move important results (like the additional baselines) into the main text, and update the text in a future version of our revised manuscript.

# 1 Novelty and generalizability [ye6c, zKep, PDvh]
In our revised manuscript, we will further emphasize that our work’s novelty does not only arise from specific loss function or methodological changes **[ye6zc-W1]**, but we show that multimodal supervision encourages more robust PPG foundation models during pre-training, while only PPG is used during inference. This allows performant generalization from clinical pre-training data to downstream field-like data (Dalia, WildPPG), and from ICU patients to young people included in lab studies (WESAD [age 27+-2.4 years], EEVR [age 23.01 +- 4.02]). The practical implications of this can not be understated, especially given the recent result that unimodal clinical PPG pre-training limits generalization to field-like data [1].

## Clinical pre-training generalization to downstream field-like datasets
Specifically, one of the main conclusions in the PulsePPG paper [1] is that “This suggests that exposure to real-world variability enables the model to learn fine-grained representations, making it more adaptable across tasks.”. This is an important result indicating that pre-training a model with clinical data does not allow it to easily generalize to real-world data. Given that large-scale field-like datasets are closed-source, our results are highly encouraging and suggest that PPG foundation models can generalize from clinical to real-world datasets as long as they make use of the inherent multimodality of many clinical datasets. Moreover, although **[zKep-W3,5]** points out that there are other datasets we can include to build a more comprehensive pretraining corpus and that the total number of data segments is comparable to PaPaGei, we want to emphasize that the reason we retain the same number of data segments with 3x fewer subjects and a single dataset is because we do not remove any noisy PPG segments from our pre-training dataset. In light of this, our performance improvements indicate that utilizing the available multimodal signals in clinical datasets allows our model to outperform 1) models pre-trained on field-like data 2) models pre-trained on multiple datasets 3) models pre-trained with more and a larger variety of subjects.

---

> ### Author Response · Authors · 2025-11-25
> **General response 2**
>
> ## Types of PPG foundation model generalization
> This also ties into **[PDvh]**’s point that the multimodal signals we use “... are closely correlated with several downstream tasks.”. Fundamentally, foundation models are pre-trained on large-scale data such that they form a base for a wide variety of tasks. In case of a PPG foundational model, the variety in tasks are:
> 1. Prediction labels: The exact downstream task that is evaluated, i.e. stress, valence, heart rate, blood pressure, etc.
> 2. Data distributions: PPG signals from either different locations on the body, from different wavelengths, or from different types of recording settings (clinical vs lab vs field)
> 3. People: Generalization to unseen demographics/groups of people
>
>
> Given these three dimensions, we emphasize that label generalization is not the only type of desirable generalization for PPG foundation models. Moreover, any downstream dataset/task we could find that involves PPG will in some way be related to ECG/RESP or other potential metrics as well. This is largely because, as **[PDvh-S1]** put it “... health is inherently multi-modal …”, and most PPG datasets obtain measures that directly (heart rate, blood pressure) or indirectly (stress, valence) relate to a person’s heart. Utilizing high-quality multimodal signals from clinical datasets that enrich embeddings with more heart-related information is thus a natural improvement. The improvement may seem obvious, but focusing only on this type of generalization diminishes the importance of the latter two types of generalization, which are especially important given the highly unique data we use to pre-train the model. Namely, as we discuss above, we show that our model, although pre-trained on clinical data, generalizes to completely different data distributions. Lastly, both through results on data obtained from young people (EEVR and WESAD) and with our within-subject evaluation framework we show that our model generalizes from a specific group of people (people in the ICU) to very different distributions of people. So even though the metrics are related to downstream tasks, some of our main contributions relate to the other two types of generalization as well.
>
> # 2 Expanding baseline comparisons [ye6c, zKep, PDvh]
> To address comments from **[zKep-W1, PDvh-W2]** that we should add comparisons against supplemental baselines, we have added evaluations for BYOL, SimCLR, and PulsePPG to Appendix F. Our results show that all baselines on average underperform our proposed model for both classification and regression tasks. The only two tasks that baselines outperform our model at are the macro F-1 score for valence classification (PulsePPG F-1: 0.54, ACC: 0.6, AUC: 0.57 vs Ours F-1: 0.53, ACC: 0.62, AUC: 0.58) and Systolic blood pressure regression for the VitalVideos dataset (PulsePPG MAE: 15.2 vs Ours MAE: 15.9). An important sidenote to these results is that we use PulsePPG’s pre-trained weights, which are obtained after pre-training on a large-scale closed-source field-like dataset. In contrast, we use a relatively small publicly available clinical dataset.
>
> Moreover, after implementing and evaluating the BYOL and SimCLR baselines, we decided to, similar to both models, perform contrastive learning on a projection of the embeddings, and have added a comparison in Appendix I. Given the improved results using a projector, all of our updated results use the projector. We want to thank the reviewers for pushing for further baseline comparisons **[zKep-W1,Q2, PDvh-W2]**, and a more stringent checkpoint selection process **[ye6c-W3]** because these changes have led to further improvements in our model’s downstream performance and more fair comparisons.

---

> > ### Author Response · Authors · 2025-11-25
> > **General response 3**
> >
> > # 3: Robustness of multimodal metrics, and using metrics over raw signals [ye6c, zKep, PDvh]
> > In light of comments about the robustness of the metric estimation in 10s windows **[ye6c-W2]**, metric estimation during PPG quality deterioration **[zKep-W2]**, and the general contribution of each multimodal signal on downstream performance **[PDvh-W5]**, we have emphasized more clearly in the text how our pre-processing pipeline ensures the multimodal metrics are robust and added an additional ablation study that ablates the use of each modality in Appendix G.
> >
> > We specifically chose to use metrics over raw multimodal signals **[zKep-W5]** to avoid any robustness issues. The concern that ECG/RESP signals are affected whenever the PPG signal is noisy makes sense, namely although ECG and RESP are generally less noisy signals, movement, for example, will still induce some noise into the ECG and RESP signal as well as the PPG signal. Clearly, noise can be a covariate for all signals. The benefit of using metrics instead of the raw multimodal signal is that they correspond to well-known physiological processes with specific bounds on their values. Specifically, heart rate has a biophysical lower and upper limit, and although extreme heart rate values may be observed for people in the ICU, we use lower and upper limits to ensure unrealistic metric values do not affect our pre-training pipeline. Moreover, because we summarize each segment with a single value for each metric, we can easily low-pass filter the metrics across a whole session. Since we use the whole session to potentially correct noisy ECG/RESP segments, we are able to ensure that the metrics are robust for individual segments. In the revised manuscript, we will more clearly stipulate this design choice, and we will move some of the pre-processing that we described in Appendix A in the original version of the manuscript into the main text to emphasize the robustness of our pre-processing pipeline. We do agree with the reviewer that looking at raw multimodal signals is an interesting future direction **[PDvh-W4]**.
> >
> > Our additional ablation study in Appendix G further shows that using ECG or RESP both lead to improved generalization outside of the pre-training dataset even when downstream tasks are not directly related to the metrics (e.g. HR regression for Dalia and WildPPG are on par or better for a RESP-only model than all baselines). Moreover, we find that both the ECG and RESP-only models obtain good performance, that ECG-only performs best on average for regression tasks, and ECG+RESP performs the best on average for classification tasks. Since the ECG+RESP model outperforms baselines across the largest number of tasks (14/16), we select it as the most versatile and performant model.
> >
> > References: \
> > [1] Saha, M., Xu, M. A., Mao, W., Neupane, S., Rehg, J. M., & Kumar, S. (2025). Pulse-ppg: An open-source field-trained ppg foundation model for wearable applications across lab and field settings. Proceedings of the ACM on Interactive, Mobile, Wearable and Ubiquitous Technologies, 9(3), 1-35.

---

### Author Response · Authors · 2025-12-03
**Summary for the Area Chair**

Dear AC,

We thank all three reviewers for their thoughtful and thorough feedback on our paper. Incorporating their comments, clarifying a few misunderstandings, and adding a substantial set of new experiments in our rebuttal has significantly improved the paper.

Our new results more clearly highlight that multimodal pre-training with noisy clinical data improves **unimodal** downstream performance, pre-training data efficiency, and allows PPG foundation models to generalize to field-like (real-life) data and non-clinical participants. All reviewers expressed a positive view of our model, including its large performance improvements (**zKep**), its clear motivation, and multimodal physiological grounding (**ye6c, PDvh**). Moreover, all reviewers indicated, even before the rebuttal, that they “would not mind if paper is accepted” with their score.

Below is a summary of the major changes we made and how our new experiments address reviewer comments:

- **[zKep-W1, PDvh-W2] Additional baselines**: We added a table (Table 13 in the current PDF) to the main text that compares our work against SimCLR and the open-source weights of PulsePPG [1]. Our new experiments show that our method outperforms all baselines (BYOL, SimCLR PulsePPG, PaPaGei-P [2], PaPaGei-S [2]) on **13/15 tasks** (Table 2 \& Appendix F), and generalizes from clinical to field-like data by leveraging multimodal pre-training signals that are uniquely available in large-scale clinical datasets, even without pre-training on a field-like dataset. The latter was suggested as necessary by prior work [1].
- **[PDvh-W5] Relationship between metrics and downstream tasks**: Because the reviewer’s concern focused on downstream label generalization, we clarified that our original experiments already assess a broader set of generalization settings: unseen labels, unseen subjects, and unseen data types. We also emphasize that generalization across subjects and data types is also essential for PPG foundation models. To address the original comment, we show in Appendix G that respiratory-only pre-training with our proposed model outperforms all baselines on field-like heart-rate regression tasks (Dalia, WildPPG), **demonstrating that our model achieves strong downstream performance even when the pre-training task is not directly aligned with the labels**.
-  **[ye6c-W2, zKep-W2, zKep-W5] Metric robustness and motivation**: We expanded our explanation for using *derived multimodal metrics* instead of raw signals. Motion artifacts often co-occur across PPG, ECG, and respiration, and deriving metrics from ECG and respiration allows session-level filtering of noisy 10-second windows based on physiological constraints. This clarification addresses the reviewers’ concerns regarding signal noise propagation and explains the motivation behind our training design.

We also addressed, among others, the following potential misunderstandings:
-  **[PDvh-W2] Unimodal vs multimodal foundation model**: We clarified that **our model is entirely unimodal at inference** and only relies on PPG signals. Multimodal signals, that are commonly available in clinical datasets, are only used during pre-training.
-  **[zKep-W5] Data efficiency**: Although we use a similar number of pre-training segments, our approach requires \~3x fewer subjects and only one of the three datasets used by PaPaGei [2], since we can retain nearly all (including often noisy) recorded PPG segments. This further strengthens our argument for both efficiency and robustness.

In addition to these major changes, we have addressed all other reviewer comments. All updates will be reflected in the revised manuscript, which will incorporate the changes described in our rebuttal.

We appreciate your consideration. Our rebuttal provides (1) substantial new empirical evidence addressing the reviewers’ main comments, (2) clarifications highlighting the full range of PPG foundation model generalization tasks, and (3) resolutions of key misunderstandings. Combined with the reviewers’ expressed openness to acceptance, despite giving a score of 4 on the coarse scoring scale, these updates provide strong evidence that the reviewers’ assessments would likely have been more favorable if score changes had been permitted.

Thank you for your time and careful evaluation.

[1] Saha, M., Xu, M. A., Mao, W., Neupane, S., Rehg, J. M., & Kumar, S. (2025). Pulse-ppg: An open-source field-trained ppg foundation model for wearable applications across lab and field settings. Proceedings of the ACM on Interactive, Mobile, Wearable and Ubiquitous Technologies, 9(3), 1-35. \
[2] Pillai, A., Spathis, D., Kawsar, F., & Malekzadeh, M. (2024). Papagei: Open foundation models for optical physiological signals. arXiv preprint arXiv:2410.20542.

---

### Meta-Review · Area_Chair_DjXu · 2026-01-06

**Summary:**

This paper has been assessed by three knowledgeable reviewers all of whom gave it marginal rejection scores. This authors propose a physiologically grounded multimodal‑supervised pre‑training approach for PPG foundation models, showing performance improvements across a range of downstream tasks. The reviewers acknowledge the motivation, clarity of writing, and the physiological plausibility of the proposed supervision strategy, as well as the competitive results vs. work such as PaPaGei. The rebuttal improves the submission by adding new relevant baselines, clarifying the role of multimodal versus unimodal signals, and providing new experiments demonstrating generalization to field‑like datasets and to settings where pre‑training metrics do not directly align with downstream labels. These additions resolve some key reviewer concerns about fairness of comparison, metric–label alignment, and the reliability of the pre‑training signals.

**Reviewer Concerns:**

Even though rebuttal has addressed a few substantive issues, but important experimental design issues persist, including the lack of ablations isolating the contribution of each pre‑training metric, the absence of equal‑capacity baselines to disentangle model size from supervisory signal, and the continued reliance on derived rather than raw multimodal inputs. Broader generalization concerns remain as well: the model is still pre‑trained solely on MIMIC, without incorporation of other large publicly available datasets, and cross‑dataset robustness analyses are still limited. While the rebuttal strengthens the work and clarifies some misunderstandings, the paper still falls short in terms of methodological novelty, breadth of evaluation, and reproducibility. The empirical gains are promising and the direction is valuable, but unresolved concerns prevent acceptance at this point.

**Reviewer Scores:**

It is unclear to me if the scores could have been improved as a result of the discussion, but even so, it is unlikely the increases would be sufficient for this paper to move into the acceptance zone for ICLR this year.

---

### Decision · Program_Chairs · 2026-01-26

Reject